# The varied sources of faculae-forming brines in Ceres' Occator crater emplaced via hydrothermal brine effusion

J. E. C. Scully [1✉], P. M. Schenk[2], J. C. Castillo-Rogez[1], D. L. Buczkowski[3], D. A. Williams [4], J. H. Pasckert[5], K. D. Duarte[6], V. N. Romero[6], L. C. Quick[7], M. M. Sori[8], M. E. Landis[9], C. A. Raymond [1], A. Neesemann [10], B. E. Schmidt [6], H. G. Sizemore [11] & C. T. Russell[12]

Before acquiring highest-resolution data of Ceres, questions remained about the emplacement mechanism and source of Occator crater's bright faculae. Here we report that brine effusion emplaced the faculae in a brine-limited, impact-induced hydrothermal system. Impact-derived fracturing enabled brines to reach the surface. The central faculae, Cerealia and Pasola Facula, postdate the central pit, and were primarily sourced from an impact-induced melt chamber, with some contribution from a deeper, pre-existing brine reservoir. Vinalia Faculae, in the crater floor, were sourced from the laterally extensive deep reservoir only. Vinalia Faculae are comparatively thinner and display greater ballistic emplacement than the central faculae because the deep reservoir brines took a longer path to the surface and contained more gas than the shallower impact-induced melt chamber brines. Impact-derived fractures providing conduits, and mixing of impact-induced melt with deeper endogenic brines, could also allow oceanic material to reach the surfaces of other large icy bodies.

[1] Jet Propulsion Laboratory, California Institute of Technology, Pasadena, CA, USA. [2] Lunar and Planetary Institute, Houston, TX, USA. [3] Johns Hopkins University Applied Physics Laboratory, Laurel, MD, USA. [4] School of Earth and Space Exploration, Arizona State University, Tempe, AZ, USA. [5] Institute für Planetologie, University of Münster, Münster, Germany. [6] Georgia Institute of Technology, Atlanta, GA, USA. [7] NASA Goddard Space Flight Center, Greenbelt, MD, USA. [8] Lunar and Planetary Laboratory, Tucson, AZ, USA. [9] Laboratory for Atmospheric and Space Physics, University of Colorado Boulder, Boulder, CO, USA. [10] Free University of Berlin, 14195 Berlin, Germany. [11] Planetary Science Institute, Tucson, AZ, USA. [12] University of California, Los Angeles, CA, USA. ✉email: Jennifer.E.Scully@jpl.nasa.gov

awn was the first spacecraft to visit Ceres, a dwarf planet and the largest asteroid-belt object (mean radius ~470 km)[1]. Dawn explored Ceres from orbit from 2015 to 2018, using its Framing Camera (FC)[2] and additional instruments[3–5]. Ceres likely formed > 3 Myr and < 5 Myr after CAIs[6] and is partially differentiated into a rocky interior and a comparatively more volatile-rich crust[1], which is composed of rock, salts, clathrates, and ≤40% water ice[7,8]. An ancient subsurface Cerean ocean would have frozen early in the dwarf planet's evolution, and remnants of this ancient ocean could still exist as subsurface brine pockets at the base of the crust[6,7,9]. In general, Ceres' surface is ubiquitously covered by phyllosilicates[10]. In addition, Ceres displays some exceptional areas, such as Occator crater. Occator is a 92-km diameter complex crater and is one of the most well-known features on Ceres' surface because of its enigmatic bright deposits, called faculae[1,11–13]. Cerealia Facula is the central bright region, mostly located in Occator's central pit. The central pit also contains a dome named Cerealia Tholus. Pasola Facula is a bright deposit located on a ledge above the central pit, while Vinalia Faculae are in the eastern crater floor (Figs. 1a and 2a). The faculae are up to 6 times brighter than Ceres' average material, as defined by ref. [14]. They are mostly composed of sodium carbonate and ammonium chloride, consistent with the remnants of brines sourced in the subsurface that lost their liquid water component on Ceres' surface[15,16]. Hydrous sodium chloride has also been observed within Cerealia Facula and, because of its rapid dehydration timescales at Ceres' surface conditions (tens of years), suggests that at least some brines may still be present in the subsurface[17].

Using data from Dawn's prime and first extended missions (≥385 km in altitude, ≥35 m/pixel FC images), multiple studies sought to uncover the sources and formation mechanisms of Occator's faculae[18]. Flows are hypothesized to have emplaced the bright material that has a more continuous appearance (corresponding to the continuous bright material geologic unit), while the discontinuous bright material, which is comparatively diffuse, was suggested to have been ballistically emplaced[18–21]. The moderately discontinuous faculae material in Vinalia Faculae has an intermediate texture (Supplementary Fig. 1 and Supplementary Discussion, subsection Bright material).

Following the prime and first extended missions, key questions remained about the source of the faculae-forming activity and the emplacement mechanism; these were some of the motivations for Dawn's second extended mission (XM2). During XM2, low elliptical orbits provided FC images of Occator with an order of magnitude higher ground sampling distance than previously obtained: as high as ~3 m/pixel from ~35 km periapsis altitude. Here we address these key questions by using the XM2 data to analyze the geologic relationships between, and physical properties of, features in Occator, via the creation of a highest resolution (XM2-based) geologic map of Occator's interior ("Methods", subsection "Geologic mapping") (Fig. 1a, Supplementary Data 1). This geologic map provides a methodically derived and self-consistent interpretation of the data that cannot be achieved by visual inspection alone. For example, the XM2-based geologic mapping reveals that almost the entire crater interior is coated by lobate material, which has been interpreted to have been emplaced as a slurry of impact-melted water, salts in solution and blocks of unmelted silicates and salts flowed around the crater interior shortly after Occator's formation[18] ("Methods", subsection "Lobate material"). While the composition of the melted material is different (water ice versus silicate rock), Occator's lobate material is the Cerean equivalent of crater-fill impact melt rocks and melt-bearing breccias found in the floors of impact craters throughout the inner solar system[22]. The now-solidified

lobate material is comparatively rich in water ice when compared with the surrounding terrain[23], and is covered by a desiccated sublimation lag most likely ≤1-m thick[24]. Hydrated salts are more stable than water ice at the same depth, and would stay hydrated in the presence of water ice[25,26]. Thus, any hydrated salts that exist at or below the ≤1 m thick sublimation lag would stay hydrated, meaning that volume loss due to dehydration would not significantly affect the topography within Occator crater.

## Results

**Brine effusion in a hydrothermal system.** Instead of identifying one centralized source region for the faculae, the XM2 data allow us to observe numerous localized bright material point features surrounding Cerealia and Vinalia Faculae, which we map as the faint mottled bright material surface feature (Fig. 1, Supplementary Fig. 1). We also observe that faculae tend to occur within the same general regions of the crater floor as fractures, domes and mounds, and that Cerealia Facula is concentrated within, and surrounding, the central pit (Fig. 3). Occator's domes and mounds may originate from eruptive and/or frost-heave-like processes derived from the solidification and expansion of the lobate material[20,27]. Examination of terrestrial impact-derived hydrothermal deposits shows that hydrothermal deposits mainly occur in crater-fill materials, the inside and outer margin of central uplifts, the ejecta, the crater rim and in crater-lake sediments[28]. The distribution of the faint mottled bright material around Occator's central pit is analogous to the uneven distribution of mounds, which are interpreted to be hydrothermal, around the central structure of the martian crater Toro[29]. Moreover, impact-derived fracture networks are found to be key drivers of the location of impact-induced hydrothermal activity[28]. A similar process appears to occur on Ceres: the relationship between Cerealia and Vinalia Faculae and prominent fractures (Figs. 1–4) indicates that pathways to the surface for the faculae-forming brines were likely opened by the prevalent impact-induced fracturing throughout the crater[30]. Moreover, excess pressures from partial crystallization of the melt chamber could also initiate and sustain fracturing[20,31].

Hydrocode simulations predict that the Occator-forming impact would have created a hydrothermal system on water-ice-rich Ceres[32], and previous work found that the morphology of Cerealia Facula is generally consistent with terrestrial, mostly non-impact-generated, hydrothermal deposits[19]. Our aforementioned morphological observations clearly show features that were not well resolved in the pre-XM2 data (e.g., the numerous localized bright material point features), and thus allow us to more definitively confirm the hydrocode modeling predictions[32]. Therefore, we find that the faculae are hydrothermal deposits that were emplaced ballistically and as flows, originating from numerous localized brine sources throughout the crater floor (e.g., the bright material point features), rather than from one centralized source region. In addition, some of the localized bright material point features are likely to be splatter deposits from the ballistic emplacement of brines[18–21]. We name this process brine effusion, which encompasses both emplacement styles (ballistic and as flows) because effusion applies to all fluids (i.e. gases and liquids). It is also a non-genetic term that does not contain implications for the source of the brines (e.g., impact-derived only, or impact-derived with an endogenic component). We note that hydrothermal systems do not have to be at or hotter than the boiling point of water: terrestrial hydrothermal springs occur at ambient temperatures[33]. Moreover, salts are precipitated from cold springs in the Canadian Arctic that are around or below 0 °C[34]. Following very hot temperatures shortly after the

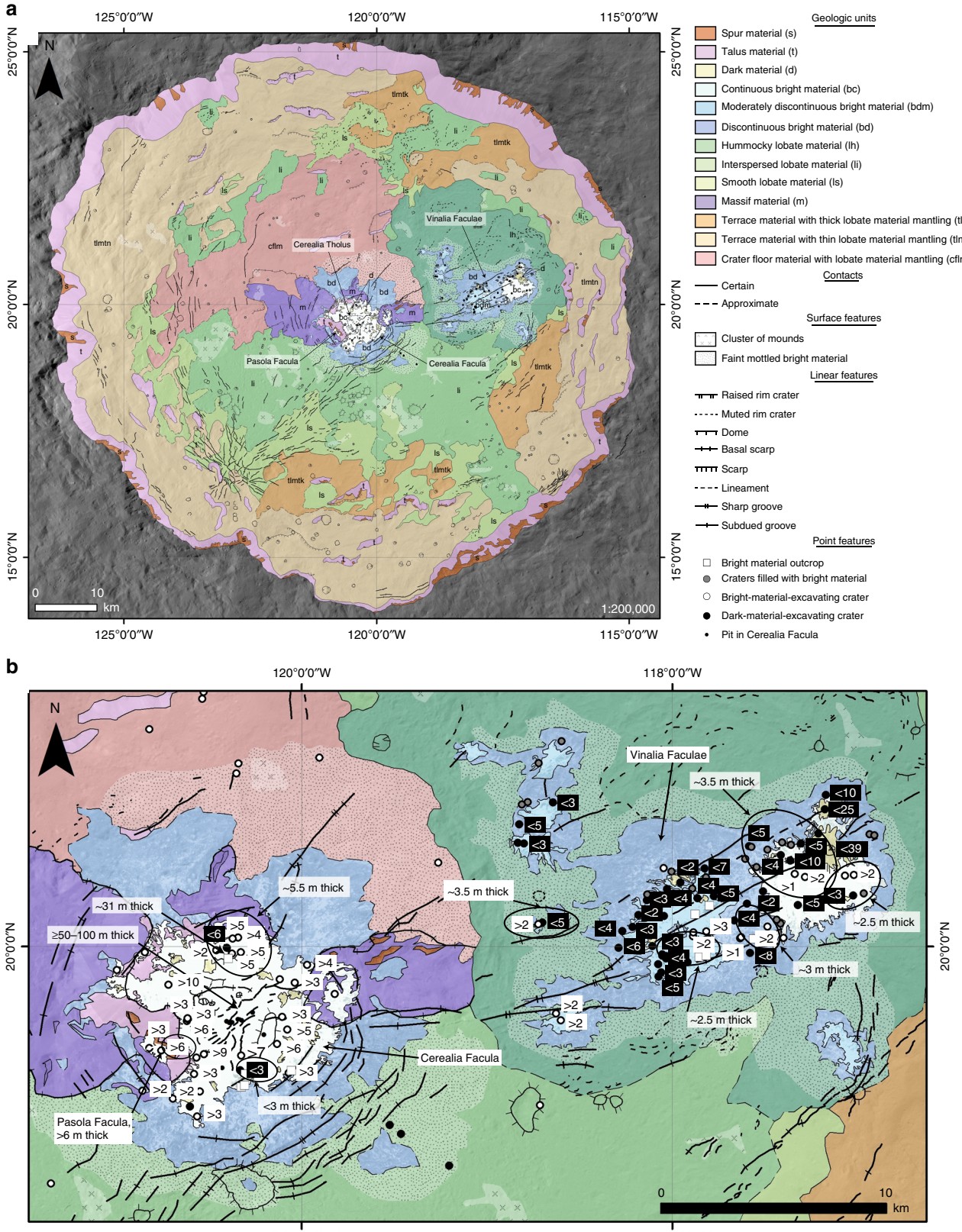

**Fig. 1 XM2-based geologic map of Occator crater's interior, and derived thicknesses. a** The geologic map is shown on the basemap at a scale of 1:200,000 and with a simple cylindrical projection. The basemap is shown with no mapping in Supplementary Fig. 4. A high-resolution JPEG, stand-alone version of the geologic map is available as Supplementary Data 1. **b** Detail of the geologic map with thicknesses of Cerealia Facula, Vinalia Faculae, and Pasola Facula derived from: superposing impact craters with dark (white text in black box) or bright (black text in white box) ejecta; an outcrop of bright material (marked by the white square symbol); and fractures on Cerealia Tholus. When thickness estimates were derived for a particular region, the region is defined with a black ellipse. Each thickness estimate is associated to an ellipse by a black arrow.

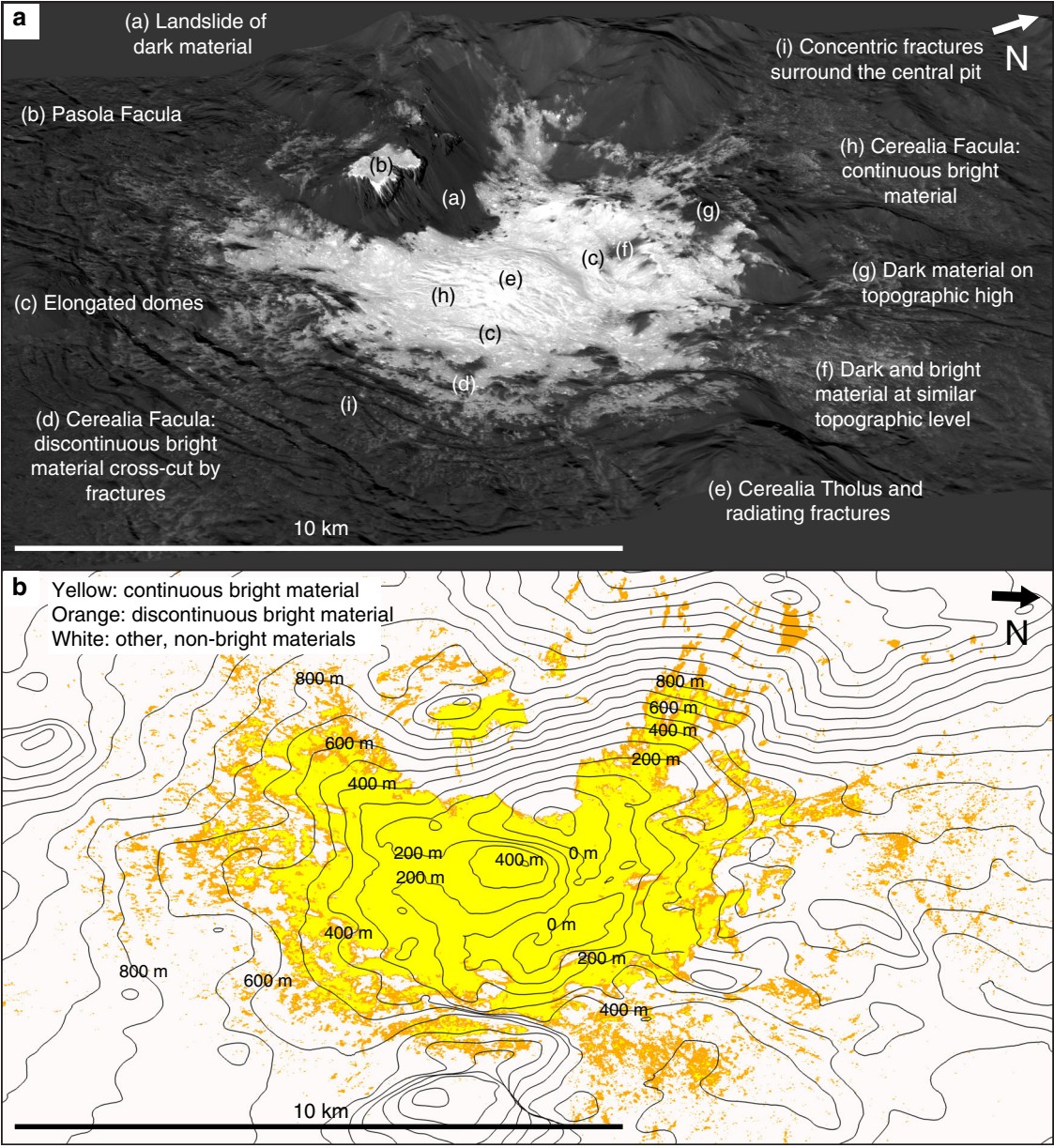

**Fig. 2 Perspective views of the central region of Occator. a** Perspective view with labels indicating key features discussed in the text, including Cerealia Facula, the central pit, Cerealia Tholus and Pasola Facula. The base mosaic is the ~10 m/pixel XM2 clear filter mosaic and is referenced to the LAMO DTM[59]. There is no vertical exaggeration. **b** Relationship between bright material and topographic lows, made using our classified version of the ~10 m/pixel XM2 clear filter mosaic. The yellow classified material approximately corresponds to the continuous bright material, the orange to the discontinuous bright material and the white to other, non-bright materials. The contours (black lines) are spaced at 100 m, and are based on the LAMO DTM. The center coordinates of both views are 19° 37′ N and 120° 24′ W.

impact, the hydrothermal systems in Occator would cool to ambient Cerean temperatures. Below the skin depth (micrometers to centimeters) at the equator, the average temperature is ~155 K[35], and slightly less at Occator, ~150 K.

Vinalia Faculae are associated with a prominent set of fractures, from which the faculae-forming brines were proposed to originate[19,21,36–38]. However, our XM2-based geologic mapping reveals that these fractures cut through the Vinalia Faculae (Figs. 1 and 4). We observe that the Vinalia Faculae fractures often broaden into pit chains coated by dark talus, and there is no clear evidence that the Vinalia Faculae bright material originated from the fractures. In contrast, we observe that landslides of bright material, originating from bright outcrops at the pits' rims, cascade down into the dark pit chains. Disaggregation of

dehydrated salts from the top ~1 m of the subsurface could form some of the loose bright material that, after becoming unstable, mass wasted into the pit chains. These observations suggest that the fractures postdate, and did not provide conduits for, the faculae-forming brines. Nevertheless, the fractures that we currently see at the surface could predate the faculae, and have formed conduits for the faculae-forming brines, if they were reactivated following faculae formation. Relatively low stresses, up to on the order of several MPa, are required to initiate fracturing on Ceres[20]. Reactivation would require even lower stresses, which makes it plausible that relatively low energy events, such as the formation of small impact craters in Occator's floor, nearby the fractures (Supplementary Fig. 2), could have reactivated the fractures to produce the currently observed cross-cutting

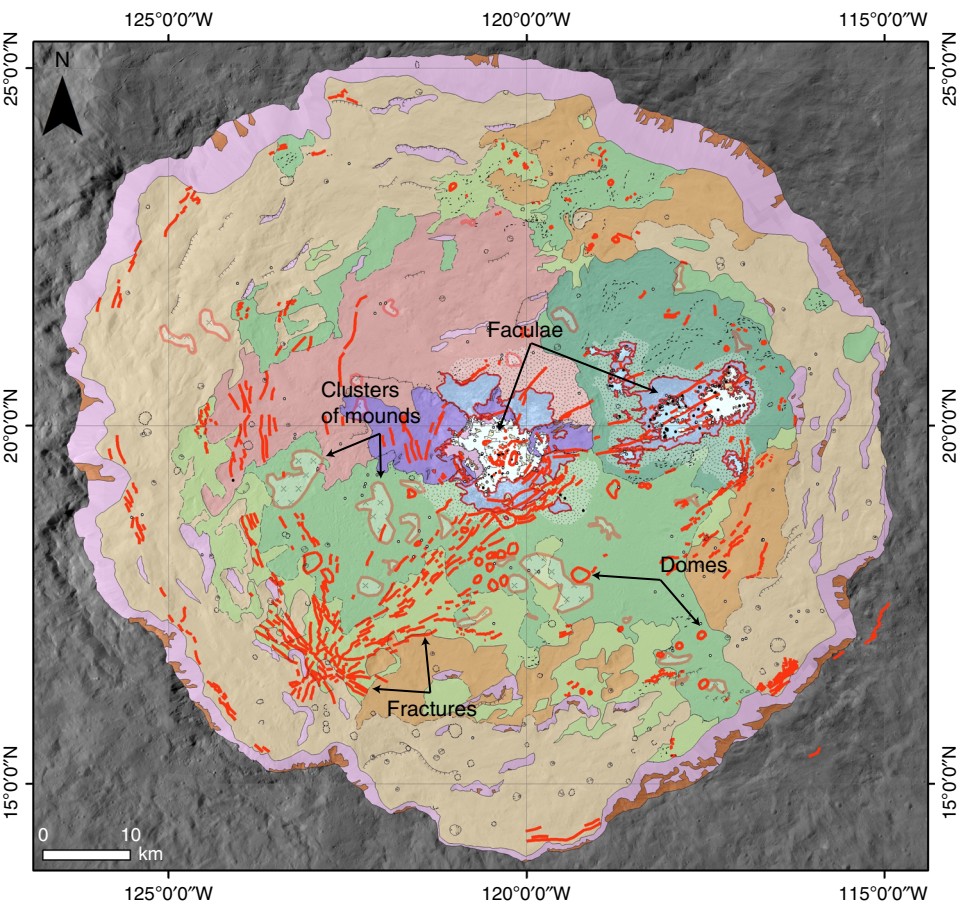

**Fig. 3 Clustering of features within Occator.** Our XM2-based geologic map (colors and symbols the same as in Fig. 1a) with fractures, faculae, domes, and mounds highlighted in red. Examples of each feature are labeled. The fractures, faculae, domes, and mounds tend to occur in the same regions of the crater floor.

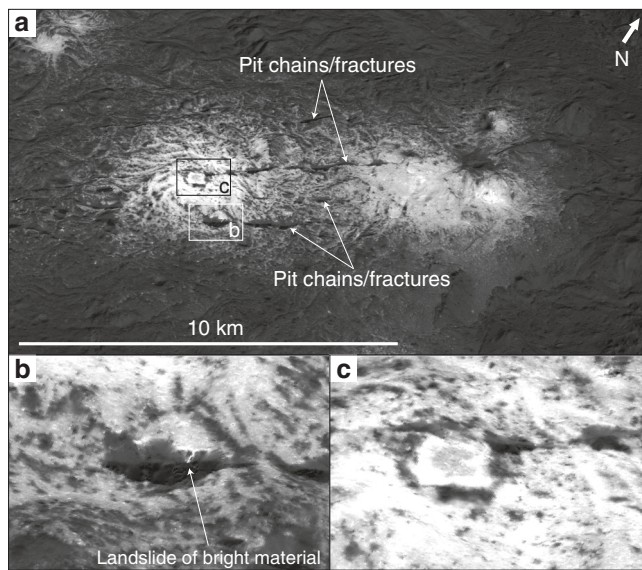

**Fig. 4 Perspective views of Vinalia Faculae. a** An overview of Vinalia Faculae. The four main pit chains/fractures that cross-cut Vinalia Faculae, and the locations of (**b**) and (**c**), are indicated. The center coordinates of this view are 20° 11′ N and 117° 34′ W. The ~10 m/pixel XM2 clear filter mosaic has 5× vertical exaggeration and to make the perspective view we referenced the mosaic to the LAMO DTM[59]. **b** A landslide of bright material cascading into a pit chain. **c** The candidate centralized source region.

relationship. Moreover, gravitational readjustment of impact-generated faults, as is observed in impact craters such as Charlevoix and Sudbury on Earth[39], could also cause reactivation. In addition, other fractures within the set, which are now buried, could have also, or alternatively, allowed the faculae-forming brines to reach the surface.

There is a candidate-centralized source region, or eruptive crater-like structure, in the center of one of the regions of Vinalia Faculae, which possibly sourced the surrounding bright material[21,40] (Fig. 4). This structure consists of a low central rise surrounded by linear depressions, and is discussed in detail in ref. [40]. The structure has straight sides, with two sides parallel to the fractures, suggesting that the weaknesses formed by the set of fractures controlled its shape. The idea that fractures could control the shape of the candidate centralized source region is analogous to the hypothesis that the shape of polygonal craters on Ceres can be attributed to subsurface fracturing[41].

In the XM2 data, there are no flow fronts clearly visible within any of the faculae, which can be explained by the buildup of ballistic deposits (in the discontinuous bright material) and by the bright material being of a sufficiently low viscosity to form a gradually sloping surface instead of a clear flow front (in the continuous bright material)[31] (Figs. 1a, 2a, Supplementary Figs. 1 and 3). In Vinalia Faculae, the moderately discontinuous faculae material may be less diffuse than the discontinuous bright material because many ballistic deposits builtup to form it. Terrestrial travertine deposits often consist of millimeter–centimeter scale layers/laminations, which were built up from successive

emplacement events[42]. Travertine is a calcite deposit formed by chemical precipitation out of solution, similar to the precipitation of the salts out of the faculae-forming brines. Thus, it is alternately possible that there may be smaller flow fronts in the faculae than can be observed in our meter-scale data.

Bright material deposits occur on the massifs surrounding the ≤1 km deep central pit: the most noteworthy is Pasola Facula, a region of continuous bright material located on a ledge that is part of the western massif (Fig. 2a). If Pasola Facula and Cerealia Facula predated central pit formation, and were originally connected[43], the pit-forming subsidence would have induced compressional stresses inside the central pit (i.e. in Cerealia Facula) and extensional stresses surrounding the central pit. We observe no contractional linear features, such as ridges or folds, in association with the central pit. However, we note that compressional stresses can occur without corresponding contractional features, and that contractional structures are often not present because they require larger differential stresses to form than extensional structures[44–46]. The only linear features inside, and surrounding, the central pit are extensional fractures/pit chains and a few scarps ("Methods", subsection "Lobate material") (Figs. 1 and 2). We do not interpret the two elongated domes within the central pit as compressional features (Figs. 1 and 2), because their morphological similarity, and location adjacent, to Cerealia Tholus leads us to interpret that they are smaller, ancillary versions of Cerealia Tholus. The XM2 data also illustrate that the border of Cerealia Facula, which is downslope of Pasola Facula, is partially obscured by landslides of dark talus. Thus, any similarities in the border pattern between the faculae are a coincidence of the deposition pattern of the dark talus material. The XM2-based geologic mapping illustrates how the dark-material superposes the bright material in many locations, indicating that the dark material is not simply a passive bystander that is either coated or missed by brine effusion. In addition, aside from a few ledge deposits at higher elevations, the bright material tends to be concentrated in topographic lows, indicating a relationship with topography (Fig. 2b). Such a relationship would not be expected if the pit collapsed after the bright materials were deposited, in which case a random distribution of bright material at various elevations would be anticipated. We also find from the XM2 data that Pasola Facula and Cerealia Facula vary in thickness: Pasola Facula is >6 m thick and Cerealia Facula ranges in thickness from <3 m at the southern side, to ~5.5 m or ~31 m at the northern side, to ≥50 m on top of Cerealia Tholus ("Methods", subsections "Faculae thicknesses from superposing impact craters" and "Cerealia Facula thickness from a bright material outcrop") (Fig. 1b). The XM2 data illustrate that there is no dark material at the base of the ~50–100 m deep[40] radiating fractures on top of Cerealia Tholus (Fig. 2a), indicating that the continuous bright material on the uppermost parts of the tholus is ≥50 m thick (Fig. 1b).

Based on all of the aforementioned evidence, we interpret that Pasola Facula was not emplaced simultaneously with, nor originally connected to, Cerealia Facula, and that the vast majority of the faculae were not emplaced prior to central pit formation. Instead, within Occator's hydrothermal system, the formation of the bright material on the massifs (such as Pasola Facula) can be explained by prevailing hydrologic gradients in the area and/or the transport of hydrothermal fluids along the prevalent fractures formed by the impact and pit collapse[30] (which may have remained open because of gravitational readjustment of impact-generated faults[39]). It is not possible to gain fine-scale resolution in the model ages derived for the faculae from crater size frequency distributions: statistical errors of Cerean model ages are sometimes as low as a few hundreds of

thousands of years, but they are typically on the order of a few millions of years or more, and do not include the larger, unquantifiable, chronology calibration errors[47]. Thus, it is plausible that there could be at least a few hundreds of thousands of years separating the emplacement of different parts of the faculae. Consequently, we interpret that the similarities in reflectance and age between Pasola Facula and Cerealia Facula[43] are because it is material with the same composition[15–17] that was emplaced from multiple sources in the same region over a similar, but not necessarily simultaneous, period of time.

Ahuna Mons, Ceres' solitary mountain that is interpreted to be an extrusive volcanic dome, is surrounded by a clear termination scarp at its base[48]. In contrast, there is only a subtle basal scarp around part of Cerealia Tholus' base (Figs. 1 and 2). The subtle basal scarp is possibly suggestive of an intrusive origin for Cerealia Tholus, such as formation via frost-heave-like processes[27], or formation as a laccolith/from volume expansion of a volatile reservoir[19]. Alternatively, if Cerealia Tholus formed extrusively, the subtle basal scarp could be attributed to the material having a sufficiently low viscosity to form a gradually sloping surface instead of a clear termination scarp, or venting of ice and gas could result in the erasure of a clear termination scarp[31]. In order to form extrusively, Cerealia Tholus would have originated from brines that increased in viscosity while relaxing into a domical shape[20,21].

**Faculae formation in a brine-limited system.** Based on the pre-XM2 data, many patches of dark material within Cerealia Facula were interpreted as topographic highs around which the faculae-forming brines flowed[19]. Our XM2-based geologic mapping shows that this relationship holds for some regions. However, many of the areas of dark material within Cerealia Facula occur at the same level as the bright material (Fig. 2a). Therefore, we interpret that the availability of the faculae-forming brines often controlled bright material emplacement, and that sufficient amounts of the faculae-forming brines were not always available to completely coat the surface.

Cerealia Facula ranges in thickness from <3 m at the southern side, to ~5.5 m or ~31 m at the northern side, to ≥50 m on top of Cerealia Tholus. Pasola Facula is >6-m thick. Vinalia Faculae have a consistent thickness of only ~2–3 m ("Methods", subsections "Faculae thicknesses from superposing impact craters" and "Cerealia Facula thickness from a bright material outcrop") (Fig. 1b). Based on these thicknesses and the areas derived from our geologic map, we find that the central faculae consist of ~11 km³ of material (using an average thickness of 40 m for Cerealia Facula and 10 m for Pasola Facula), while Vinalia Faculae consist of only ~0.6 km³ of material (using an average thickness of 2.5 m). Thus, Vinalia Faculae appear to have been significantly more brine-limited than the central faculae. Moreover, we identify 20 circular depressions (interpreted as impact craters) that are partially infilled by bright material in Vinalia Faculae. While there are many bright, circular features in Cerealia Facula, there are no partially infilled impact craters. The lack of partially infilled impact craters in Cerealia Facula is consistent with Vinalia Faculae being comparatively thinner and more brine limited, because the thicker Cerealia-Facula-forming brines would have filled in any pre-existing impact craters ("Methods", subsection "Faculae thicknesses from partially infilled impact craters").

**Different sources for central faculae and Vinalia Faculae.** Hydrocode and thermal modeling of the Occator-forming impact predict that impact-melted water ice mixed with salts, both from

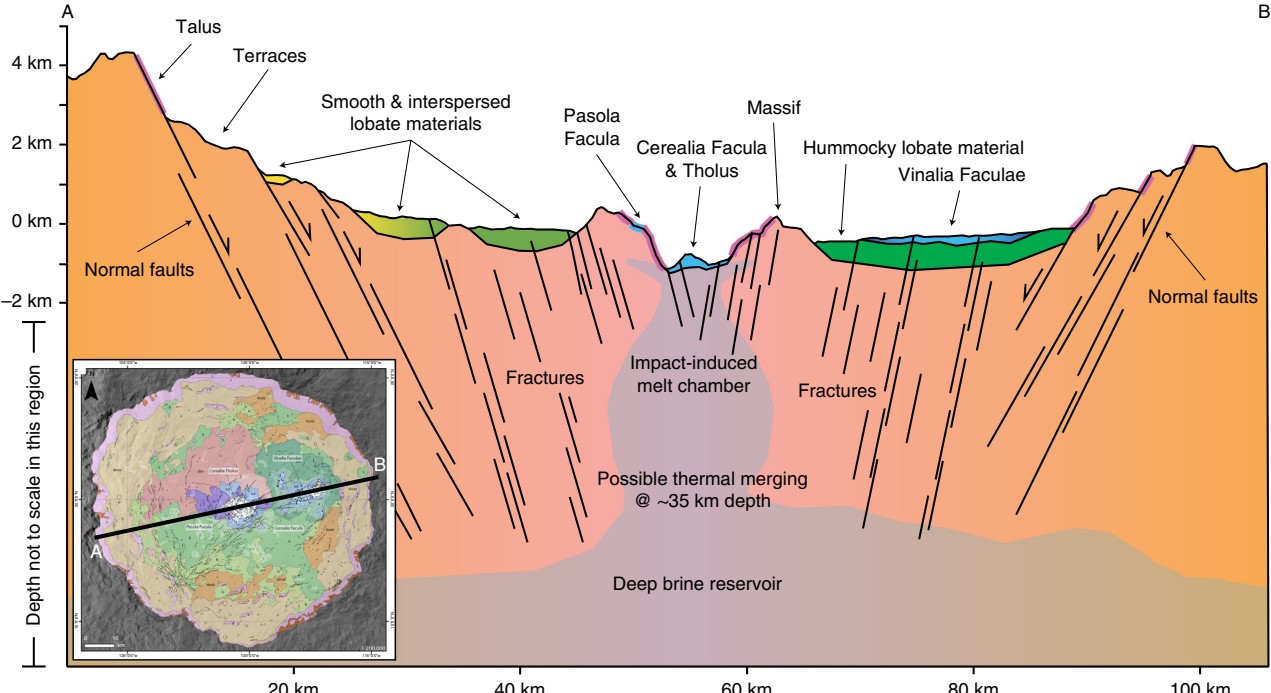

**Fig. 5 Cross section through Occator crater, including schematic subsurface structure.** The colors shown in this figure correspond to the geologic units (as defined in Fig. 1a) and key features are labeled. This figure does not illustrate a particular time-step in Occator's evolution. Impact-derived fractures form conduits to source the faculae-forming brines, and the impact-induced melt chamber thermally connects to the deep brine reservoir. The sizes and depths of the impact-induced melt chamber and deep brine reservoir are schematic, and are based on refs. [9, 30, 32, 49]. We do not show the warping of the lobate material by the formation of the central pit in the cross section, because it is out of the plane, which was chosen to show the key features in the crater. The profile is taken from the LAMO DTM[59], and the line of the profile is shown in the inset image.

Ceres' crust, would form a briny melt chamber in the center of the crater, which would be roughly 20 km in diameter and extend from the shallow subsurface down to ~20 km[30,32,49] (Fig. 5). The central faculae in our geologic map form a roughly 20 km circle, thus fully encompassing this impact-induced melt chamber's extent. From this consistency between modeling and mapping results, we infer that both Cerealia Facula and Pasola Facula were locally fed by brines sourced in the impact-induced melt chamber. Some solid material (e.g., silicates, which the impact would not be hot enough to melt[32]) would likely be mixed into this melt chamber and, over time, solidification would increase the solid fraction of the impact-induced melt chamber[32,49].

In contrast, Vinalia Faculae are located far from the crater center: ~20 km separates the centers of Cerealia and Vinalia Faculae, while the farthest edge of Vinalia Faculae is ~30 km from the crater center. Hydrocode modeling indicates that the impact-induced melt chamber is only ~20 km in diameter[30,32,49], thus making it an unlikely source for the Vinalia Faculae-forming brines. Instead, Vinalia Faculae could be sourced from a deep, long-lived brine reservoir, which has been suggested to be present at the base of the crust (~35-km deep) on the basis of topographic analyses[7] and is supported by thermal modeling[9]. This deep brine reservoir would have existed prior to the Occator-forming impact and is inferred to be present on a global scale[7,9,30], although the amount of liquid may vary laterally[9] (Fig. 5). The impact-induced melt chamber would likely thermally connect to this deep brine reservoir[30,49]. Therefore, the central-faculae-forming brines primarily originated from the impact-induced melt chamber, with likely long-term contributions from the deep brine reservoir, while the Vinalia-Faculae-forming brines only originated from the deep brine reservoir.

The impact-induced melt chamber, which feeds the central faculae, is predicted to extend to much shallower depths than the

deep brine reservoir[9,20,30,32,49] (Fig. 5). In contrast, the Vinalia Faculae-forming brines, sourced in the deep brine reservoir via fractures, would take a longer, and thus likely more difficult, path to the surface than the Cerealia- and Pasola-Facula-forming brines. Consequently, the deeper source of Vinalia Faculae is consistent with Vinalia being more brine limited than the central faculae, as indicated by the relatively small thicknesses and volumes of the Vinalia Faculae in comparison to the central faculae (Fig. 1b). In addition, the sodium-carbonate-rich composition of Vinalia Faculae[15–17] constrains the composition of the deep brine reservoir, which requires a temperature of >245 K for sodium carbonate to be abundant in solution[9,20].

A shorter period of emplacement, instead of different sources, could alternatively form less voluminous deposits at Vinalia Faculae. However, relative stratigraphic relations do not provide evidence for the emplacement durations of the faculae: the central faculae and Vinalia Faculae both superpose, and are superposed by, the same geologic units (Fig. 1a) (Supplementary Discussion, subsection Bright material), meaning that relative emplacement durations cannot be derived. A set of crater-count-derived model ages suggest that Cerealia Facula began to form ~8 million years ago, while formation of Vinalia Faculae began ~4 million years ago[43] ("Methods", subsection "Crater-count-derived model ages"). While the model ages provide approximate ages for the faculae, and there is some evidence for possible local reactivation/resurfacing on Cerealia Facula ~1–2 million years ago[43], the model ages cannot precisely quantify the emplacement durations of the faculae. Moreover, the small count areas used for the faculae, which have low crater densities and less large craters, are more susceptible to stochastic cratering variability, contamination by secondary craters and degradation/erasure of small craters than larger areas[40,47]. In addition, the frequently diffuse nature of the deposits make absolute age dating of the faculae notoriously

difficult, and prone to relatively large uncertainties[40,47]. Thus, while a shorter period of emplacement for Vinalia Faculae cannot be entirely ruled out, there is no clear evidence supporting this possibility.

In our geologic map, the percentage of the total area encompassed by the continuous bright material in Cerealia Facula is 2.5 times that of the percentage of the total area encompassed by the continuous bright material in Vinalia Faculae. This is consistent with there being more ballistic emplacement at Vinalia Faculae[18,20,21], which provides surficial evidence for the gaseous content of the faculae-forming brines. The shallower ($\leq$35 km deep[9]), impact-induced melt chamber would be under lower pressure. Consequently, volatiles would be partially exsolved from these brines before they were emplaced onto the surface, resulting in comparatively less ballistic emplacement. In contrast, the deep brine reservoir ($\geq$35 km deep[9]) would be under greater pressures, keeping more volatiles in solution until they neared the surface, resulting in comparatively more ballistic emplacement at Vinalia Faculae, as observed.

It is likely that Cerealia Facula and Pasola Facula formed in the center of the crater because the impact-induced melt chamber (with likely long-term contributions from the deep brine reservoir) provided a shallow, readily available brine source. It is more difficult to explain why Vinalia Faculae formed in the eastern crater floor and no other faculae formed elsewhere in the crater floor. While there is a relationship between the occurrence of faculae and fractures in Occator's floor (Fig. 3), faculae are not associated with all of the prominent fractures. Most notably, there are no bright deposits like Vinalia Faculae associated with the cluster of fractures that form a radial pattern in the southwestern part of Occator's floor. This cluster of fractures occurs at the boundary between the smooth lobate material and the terrace material with thin lobate mantling (Fig. 1a). Perhaps the terrace material in this region provided a more competent barrier (in comparison to the lobate materials) through which the faculae-forming brines could not flow. Alternatively, perhaps the fractures in this region were configured in a manner that did not provide a viable pathway to the surface. While our current data and models do not provide a definitive explanation for why all of the prominent fractures in Occator do not source faculae, this observation is consistent with our earlier interpretation that the system was brine limited.

## Discussion

While it is possible that alternative factors, such as period of emplacement, could control the differing volumes of the central faculae and Vinalia Faculae, the varied sources hypothesis can explain both the different volumes and different dominant emplacement styles between the faculae. Thus, we conclude that the central faculae (Cerealia Facula and Pasola Facula) were sourced in an impact-induced melt chamber, with a contribution from the deep brine reservoir, while the Vinalia Faculae were sourced by the deep brine reservoir alone. Occator crater formed ~22 Myr ago[47] and the faculae could have formed as recently as a few millions of years ago[38,43,50] (based on the lunar derived chronology model[51]). However, the impact-induced melt chamber could have only existed for ~12 Myr without a contribution from the deep brine reservoir[49]. Thus, the role the deep brine reservoir played in the formation of all of the faculae explains how the faculae formed many millions of years after the impact-induced melt chamber would have cooled and solidified.

Here we show that geologic mapping of surficial deposits can be used in conjunction with modeling studies to make inferences about subsurface structure: we identify different, sometimes connected, sources for the faculae, and find that fractures formed

by the impact[30], and from partial crystallization of the melt chamber[20,31], allowed the faculae-forming brines to reach the surface. The melt chamber and fractures formed by the Occator impact reached, and mixed with, deep brine reservoirs, and consequently sourced materials that would otherwise have not reached the surface. Cryovolcanism on the icy satellites of the outer solar system (discussed by, for example, refs. [52–55]) can be formed by excess pressures from crystallization of reservoirs[31]. Alternatively, it is also possible that similar processes to those observed at Occator could occur on the icy satellites and other large icy bodies (e.g., dwarf planets and large KBOs): impact-derived fractures could form conduits, and merging of impact-derived and pre-existing reservoirs, including deep oceans, could mix materials originating from different depths, thus enabling their emplacement on the surface and detection/investigation by space missions.

## Methods

**Geologic mapping.** Geologic maps of Occator have been published using data from Dawn's prime and first extended missions[36,37,47,50,56,57]. Here we present a geologic map of the interior of Occator crater made using the FC images obtained during XM2, specifically the low elliptical phase, which have an order of magnitude higher ground sampling distance than previous data. Our basemap is the XM2 clear filter FC mosaic[58] (Supplementary Fig. 4). It is a ~3 m/pixel controlled mosaic, and is orthorectified onto the Low Altitude Mapping Orbit digital terrain model (LAMO DTM)[59]. Some regions of the basemap were imaged at >3 m/pixel, and thus these regions were interpolated to ~3 m/pixel. The southernmost and westernmost parts of Occator's interior are outside of our basemap. In these areas we supplemented our basemap with a XM2 clear filter FC controlled mosaic (~10 m/pixel) and the LAMO clear filter FC controlled mosaic (~35 m/pixel)[60], which is orthorectified onto the LAMO DTM[59].

The boundary of our geologic map is the rim of Occator. We mapped the entire crater interior at 1:50,000 and the faculae at 1:10,000. We used a combination of 2D mapping in ESRI ArcMap and 3D mapping in ESRI ArcScene[61]. By referencing the basemap and supplementary datasets to the LAMO DTM in ArcScene, we were able to view the data in 3D perspective views. We first created a rough map using the ArcScene 3D perspective view as a base, before transferring the mapping into the 2D view in ArcMap for refinement. Creating our geologic map using both 2D and 3D views facilitated greater insights into the placement of contacts, stratigraphic relations etc. than 2D mapping alone. To account for the large brightness differences between the faculae and surrounding terrains, we varied the standard deviation stretch of the basemap when mapping (Supplementary Discussion, subsection Description of Map Units). Our mapping approach was informed by United States Geologic Survey (USGS) practices for the definition of units, placement of contacts, choice of symbol types, etc. However, the necessity of creating the geologic map during an active mission meant that we did not create a USGS Science Investigations Map, which is typically produced over many years after a mission has ended[62].

**Lobate material and central pit.** The lobate material was emplaced as a slurry of impact-melted water, salts in solution and blocks of unmelted silicates and salts flowed around the crater interior soon after Occator's formation[18]. Based on the timescale for conductive cooling ($t = (L^2)/\kappa$), we find that this timescale was in the range of a few 1000–100,000 years. $L$ is the thickness of the flow, and we use $\kappa = 1 \times 10^{-6}$ m$^2$/s as the thermal diffusivity of a water-rich flow on Ceres. Thus, water-ice-rich lobate flows, such as the ~200–600 m thick lobate materials in Occator, would cool and solidify within a few 1000–10,000s years. Note that if the lobate materials are briny/salt rich, $\kappa = 1 \times 10^{-7}$ m$^2$/s, solidification would occur on the order of 10,000 years (for $L = 200$ m) to 100,000 years (for $L = 600$ m).

The slurry may have suspended the blocks of unmelted silicates and salts, in a similar process to a debris flow. While the thick sheet and smaller, pond-like deposits of lobate material were observed in pre-XM2 data, the thinner veneer of lobate material that coats the majority of the terraces and the crater floor is only clearly visible in the XM2 data (Fig. 1a). Thus, our XM2-based geologic map illustrates that almost the entire crater interior is coated by lobate material. The thin veneer of lobate material often forms a cap that breaks off at the steeply sloping terrace edges, which are covered in talus (Supplementary Fig. 5a). The geometry of such cliffs implies a lower limit on material strength that is consistent with compositional constraints inferred from other techniques (Supplementary Methods and Supplementary Fig. 13). The XM2 data illustrate that the lobate material often flows under the control of the underlying topography, and that different lobate material flows often superpose one another[40] (Supplementary Figs. 5b and Supplementary Discussion, subsection Lobate material).

A ~500 m thick ridge of lobate material cross-cuts an impact crater in the western area of the mantled terraces, indicating that the mantling of the terraces and crater floor can be thick in places (Supplementary Fig. 6d). However, the

mantling appears to generally be rather thin. We find that more craters have excavated boulders in the mantled crater floor material and mantled terrace material than in the lobate material (Supplementary Fig. 7b). It is possible that boulders formed from the comparatively water-ice-rich lobate material are preferentially removed because of thermal breakdown. However, this observation is also consistent with the mantled crater floor and terraces being covered in relatively thin lobate material mantling, through which the impact craters excavated to the underlying, more competent terraces, which sourced the boulders. Using an excavation depth to diameter ratio of >0.08[63,64], and the diameters of craters that excavate boulders, we find that the lobate material mantling in these regions is typically up to a few tens to a few hundreds of meters thick. We also mapped all of the impact craters >400 m in diameter that occur within Occator, and classified their rims as either raised or muted (Fig. 1a). We mapped craters with distinctive features on a separate map, such as craters that are cross-cut by fractures and craters that excavate boulders (Supplementary Fig. 7b). All types of impact craters tend to be concentrated in the mantled terraces and crater floor, and are rarely found in the faculae, which is consistent with the young model ages derived for the faculae[43] ("Methods", subsection "Crater-count-derived model ages"). The majority of the >400-m diameter craters are concentrated in the southern part of Occator's interior, which is consistent with the location of an ENE-WSW-trending secondary crater cluster[47]. Thus, the enhancement of >400-m impact craters in this region is probably caused by secondary contamination[47].

Using surface texture, we classify the lobate material into different sub-units, of which smooth lobate material and hummocky lobate material are endmembers (Supplementary Discussion, subsection Lobate material and Supplementary Figs. 8a–c). We define an intermediate sub-unit based on the detailed textural information in the XM2 data: smooth lobate material interspersed with striations and knobs (often referred to as the interspersed lobate material for brevity). Striations are frequently observed at the ends of lobate flows (Supplementary Fig. 8d), are indicative of the flow direction (Supplementary Fig. 9) and formed as the material flowed shortly before solidification. The majority of the knobs are in the lobate material (classified as domes or mounds in our geologic map (Supplementary Discussion, subsection Lobate material and Fig. 1a)), which is consistent with all of the possible dome and mound formation mechanisms: (a) eruptive and/or frost-heave-like processes derived from the solidification and expansion of the lobate material[27], (b) pinnacles around which the lobate material flowed, and (c) entrained blocks of unmelted silicates and salts.

Vinalia Faculae are located in the hummocky lobate material, which is proposed to form via inflation resulting from the injection of an ice/salt intrusion[36]. We investigate the possibility that the faculae-forming brine effusion resulted from the solidification and expansion of the water-ice-rich lobate material, similar to the formation mechanism proposed for the domes and mounds[27]. Water-ice-rich lobate flows, such as those in Occator (~200–600 m thick), would cool and solidify in the range of a few 1000–100,000s years. However, this timescale is orders of magnitude shorter than the time difference between the formation of the lobate material and the faculae, as estimated from crater counts[43,47] ("Methods", subsection "Crater-count-derived model ages"). Thus, the lobate material likely solidified long before faculae formation, making it an implausible source for the faculae-forming brine effusion.

Cerealia Facula coats the majority of the central pit, which is surrounded by concentric and radial fractures that formed as the pit subsided[19,36,37] (Fig. 2a). While much of the stress arising from pit formation was accommodated by fracture formation, perspective views of the XM2 data illustrate how pit formation warped the northern part of the lobate material sheet (Supplementary Fig. 10). Consequently, we interpret that the central pit formed relatively early in the crater's evolution[19], prior to complete lobate material solidification. The fractures concentric to the central pit sometimes cross-cut parts of the Cerealia Facula discontinuous bright material[19,36–38] (Fig. 2a). Thus, in keeping with our proposition that faculae deposition need not be simultaneous (Main Text), the parts of the discontinuous bright material cross-cut by the fractures could have formed prior to the central pit, while the majority of brine effusion occurred after central pit formation. However, because fracture reactivation does not require high stresses on Ceres (Main Text), we cannot discount the possibility that the aforementioned cross-cutting relationship is due to reactivation of the concentric fractures after pit formation. In this case, the parts of the discontinuous bright material cross-cut by the fractures could have formed prior to the central pit, and all brine effusion could have occurred prior to central pit formation.

**Faculae thicknesses from superposing impact craters**. When formation of the faculae had mostly ceased, the chief processes within the crater were localized modification by mass wasting and impacts[37]. Many bright and dark patches were identified within the faculae in the pre-XM2 data, but because of the resolution of the data it was often difficult to positively identify whether they were impact craters and their ejecta. The XM2 data have allowed for the identification of ~160 total bright and dark impact craters and their ejecta, some of which superpose the faculae (Fig. 1b). Here we use the superposing impact craters with dark or bright ejecta to estimate localized thicknesses throughout Cerealia Facula, Pasola Facula, and Vinalia Faculae.

There are crater-like features in Cerealia Facula that we map as pits (Supplementary Fig. 11a). They do not have regular bowl shapes and may be vents

through which brines were ballistically emplaced, because ballistic eruptions can be easily driven by less than 1% volatiles in Ceres' low gravity environment[20,21]. Such endogenic pits could be formed by release of volatiles during cooling of the crater[40], in a manner reminiscent of the formation of pitted terrain on Mars, Vesta, and Ceres by degassing of impact-heated volatile-bearing materials[65–68]. While the endogenic pits share some morphological similarities with the pitted terrain (such as a lack of raised rims and irregular shapes), Occator's endogenic pits are more isolated and coalesce less than typical pitted terrain. To ensure we used impact craters for our thickness estimates and not other depressions, such as these endogenic pits, we only used features with regular bowl shapes, ejecta, and raised rims. We use all three criteria to identify impact craters, in order to lower the possibility of false detections. For example, endogenic pits could be surrounded by an ejecta-like deposit, but are less likely to have regular bowl shapes. Nevertheless, the difficulty in definitively identifying all impact craters from endogenic pits will likely contribute some unquantifiable errors to the derivation of model ages by the studies discussed in the "Methods" (subsection "Crater-count-derived model ages").

We divided the impact craters into two classes: bright and dark. In order to make the craters clearly visible, we mapped them with the following standard deviation stretches on the basemap: bright craters in Cerealia Facula and Pasola Facula ($n = 20$), bright craters in Vinalia Faculae ($n = 15$), and dark craters in all faculae ($n = 7$). All of the craters we used are well within the size range of simple craters on Ceres, because the simple to complex transition occurs at ~7.5–12 km[51]. Thus, we used the excavation depth for Barringer crater (a simple crater), which is >0.08 times the final rim diameter[63,64]. Craters that excavate bright material yield minimum faculae thicknesses, while craters that excavate dark material yield maximum faculae thicknesses.

Combining adjacent minimum and maximum thicknesses allows us to estimate actual thickness in the localized area. We display the thickness estimates on a dedicated version of our geologic map (Fig. 1b). We find that the thickness estimates for Vinalia Faculae cluster around 2–3 m (consistent with previous studies[38]), while the thickness of Cerealia Facula varies: the material is thinner around the edges (<3 m or ~5.5 m thick in specific locations), and thicker on the top of Cerealia Tholus (≥50 m[40]) (main text). The greater abundance of dark-material-excavating craters within Vinalia Faculae (38, based on our geologic map) than Cerealia Facula (7, based on our geologic map) are also consistent with Vinalia Faculae being thinner than Cerealia Facula. In addition, the majority of the craters that excavate bright material are in the continuous bright material rather than in the moderately discontinuous and discontinuous bright materials, which is consistent with the moderately discontinuous and discontinuous bright materials being more diffuse and thinner.

**Cerealia Facula thickness from a bright material outcrop**. Outcrops of bright material, which are only clearly resolved in the XM2 data, occur along a scarp in Cerealia Facula and at the rims of the pit chains that cross-cut both faculae (Fig. 1b). We map the approximate centers of these outcrops as point features. We measured the scarp in Cerealia Facula at four locations to find the average thickness, using a standard deviation stretch of $n = 15$. In order to calculate the thickness from these measurements, we assumed the faces we measured were the hypotenuses of right-angled triangles that contain two 45° angles. Thus, we find the thickness of the material using the formula $a = h \sin A$, where $a$ = actual thickness (m), $h$ = thickness measurement of outcrop face (m) and $A = 45°$. We add the resulting thickness estimate (~31 m) to our map of localized faculae thickness (Fig. 1b), where it can be seen that there is a gradient in thickness across Cerealia Facula: we find the edge of the continuous bright material is <3 m thick, the mid region is ~5.5 m to ~31 m thick, and the crest of Cerealia Tholus is ≥50 m thick.

**Faculae thicknesses from partially infilled impact craters**. We identify 20 circular depressions partially infilled by bright material in Vinalia Faculae, and no such features in Cerealia Facula. We interpret the circular depressions as impact craters with a dark rim and bright interior deposit because they appear to be roughly circular in shape and have clear rims (Supplementary Fig. 11b). There are three possible explanations for the presence of partially infilled impact craters in Vinalia Faculae, and none in Cerealia Facula: (a) the Vinalia Faculae are thinner deposits, which do not completely bury the pre-existing impact craters, while the thicker Cerealia Facula do completely bury them; (b) there was comparatively more emplacement via flows in Cerealia Facula, which completely buried pre-existing impact craters, while there was more ballistic emplacement at Vinalia Faculae, which emplaced material more diffusely and only partially infilled the pre-existing impact craters, and (c) Vinalia Faculae were emplaced more recently than Cerealia Facula, meaning there were more pre-existing craters to be partially infilled. Crater counts suggest that there could be approximately a few million year age difference between Vinalia Faculae and Cerealia Facula[43], but a few millions of years is unlikely to be a sufficient duration to facilitate the accumulation of many more impact craters in the Vinalia Faculae region prior to the deposition of the bright material. Thus, we hypothesize that a combination of options (a) and (b) explains the presence of partially infilled impact craters in Vinalia Faculae only.

**Crater-count-derived model ages**. There are two different chronology systems in use for Ceres: the lunar derived chronology model and the asteroid-flux derived chronology model[51]. Using the lunar derived chronology model, ref. [47] find a model

age for the lobate material of ~18 Ma, while ref. [43] estimate that Cerealia Facula and Vinalia Faculae formed between ~1–8 million years ago (Supplementary Fig. 12): Cerealia Facula mainly formed $7.5^{+2.6}_{-1.7}$ million years ago, with possible local reactivation/resurfacing about $2.1^{+0.3}_{-0.7}$ and $1.2^{+0.4}_{-0.3}$ million years ago, while model ages for Vinalia Faculae range from $3.9^{+0.3}_{-0.3}$ to $1.7^{+0.7}_{-0.5}$ million years ago[43]. Thus, based on the results of the lunar derived chronology model, there is a ~10-17 million year time difference estimated between the formation of the lobate material and the faculae.

Currently, there are no peer-reviewed asteroid-flux derived model ages for the faculae; preliminary ages for Vinalia Faculae derived using the asteroid-flux derived chronology model indicate formation <1 million years ago[69]. Model ages derived for the lobate material based on the asteroid-flux derived chronology model range from ~1-53 Ma, based on the scaling parameters assumed[47] (Supplementary Fig. 12). The younger ages (~1–10 Ma) return reasonable fits to the crater size frequency distribution measurements, but were derived from material strengths that may be unreasonably low for Ceres. The older ages (~12–53 Ma) are derived from higher material strengths, but deviate more from the crater size frequency distribution measurements. Thus, for a specific set of scaling parameters, some of which may not be applicable to Ceres' surface, the asteroid-flux derived model suggests that the lobate materials could have formed much more recently (as recent as ~1 million years ago) than predicted by the lunar derived chronology model. However, even if the lobate materials are as young as ~1 Ma, approximately one million years is still somewhat older than the solidification timescale of the lobate material: in the range of a few 1000–100,000s years ("Methods", subsection "Lobate material"). Thus, when we compare the model ages derived from both the lunar and asteroid-flux derived chronology systems with the solidification timescale of the lobate material, it appears that the lobate material is an unlikely source for the faculae-forming brines.

## Data availability

The datasets generated during the current study are included in this published article, and in the Supplementary Information and Supplementary Data 1 (a high-resolution JPEG, stand-alone version of the geologic map). The datasets analyzed during the current study are available in the PDS Small Bodies Node repository, https://pds-smallbodies.astro.umd.edu/data_sb/missions/dawn/. XM2 data (specifically Dawn Ceres FC2 raw, XMO7) can be found at https://sbn.psi.edu/pds/resource/dawn/dwncfcL1.html. The LAMO clear filter mosaic (specifically Ceres LAMO Clear Filter Mosaics (version 2)) can be found at https://sbn.psi.edu/pds/resource/dawn/dwncfcmosaics.html. The LAMO DTM of Occator (specifically Ceres Regional DTMs and Mosaics for selected regions) can be found at https://sbn.psi.edu/pds/resource/dawn/dwncfcshape.html.

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

## Acknowledgements

Part of the research was carried out at the Jet Propulsion Laboratory, California Institute of Technology, under a contract with the National Aeronautics and Space Administration. We thank the Dawn Flight Team at JPL for the development, cruise, orbital insertion and operations of the Dawn spacecraft at Ceres. We thank the instrument teams at the Max Planck Institute (MPS), German Aerospace Center (DLR), Italian National Institute for Astrophysics (INAF), and Planetary Science Institute (PSI) for the acquisition and processing of Dawn data. The ~10 m/pixel XM2 clear filter FC controlled mosaic was made by Dawn Science Team member David P. O'Brien (PSI). © 2020. All rights reserved.

## Author contributions

J.E.C.S. led the geologic mapping, additional analyses, interpretation of the data, and preparation of the manuscript. D.L.B., D.A.W., J.H.P., K.D.D., and V.N.R. undertook geologic mapping, which was compiled together by J.E.C.S. under consultation with the aforementioned co-authors. L.C.Q. calculated the timescale for conductive cooling of the lobate material and M.M.S. undertook the cliff stress modeling. P.M.S., M.E.L., J.C.C.-R., L.C.Q., H.G.S., A.N., B.E.S., C.A.R., and C.T.R. contributed to interpretation of the data and the preparation of the manuscript.

## Competing interests

The authors declare no competing interests.
