## [Peer Review File · Nature Communications]

REVIEWERS' COMMENTS:

Reviewer #1 (Remarks to the Author):

This was an interesting and generally well written paper to read. Minor comments are provided in the attached annotated pdf while some more major comments are below. My two most critical comments are highlighted in bold below and relate to the very short presentation of the impact-generated hydrothermal system case and the talk of an impact melt chamber, which is not consistent with what we know about impact craters on Earth and other planetary bodies.

L47: The authors use the term “central pit”. This term has been used for craters on Mars and other bodies with various widely differing explanations for why there is a pit present versus a central peak. To avoid confusion I’d suggest using the term “central depression” as you haven’t really justified as to why this morphological feature is a “pit”.

L58 and elsewhere: The authors make quite a few very unequivocal statements using “was”. Given everything remains an interpretation or a hypothesis until ground-truth from surface missions happen, I would urge the authors to tone down some of these statements and use phrasing such as “have been interpreted to have formed by X” etc.

L78-81: Given the main conclusion of this paper is that Occator Crater’s faculae formed from impact-generated hydrothermal activity, I think this sentence represents a huge oversimplification of the topic. I also think that this is putting the cart before the horse and I would strongly urge the authors to lay out their observations first and then explain why they think impact-generated hydrothermal activity was the process responsible for forming the faculae.

With respect to this sentence, the hottest locations in impact craters (by far) are not central uplifts, but rather the crater-fill impact melt-bearing breccias and melt rocks, so at the very least, this needs to be reworded. In the Osinski et al. paper you reference, the various locations within and around craters where hydrothermal deposits are located is presented. I think a short summary of this is key here.

L96-99: As a suggestion, reactivation need not be from some disparate post-impact process. It is known from studies of craters on Earth that faults generated during the few seconds to minutes of the crater forming process, will gravitationally readjust over thousands to millions of years.

P102-103: I am not convinced that this is a vent feature. Do you have topographic data to show that this is comprised of a central depression surrounded by a rim, which would constitute a vent?

L108-109: Here and elsewhere the authors refer to reference #24. I read this manuscript and I find considerable overlap in terms of the conclusions with this current manuscript. In their abstract the write “Our results suggest that impact heating from the Occator forming impact provides a viable mechanism for the creation of the observed faculae”. I think the authors need to be clearer about how their work differs from this previous/recently published study, given the similar conclusions.

L206-212: I am having a hard time understanding this text and the Figure 5 and the suggestion of a “melt chamber formed by the impact”. I do not have access to

reference #19, but nowhere in reference #31 do I see mention of this and in reference #24, those authors talk about “a ‘plug’ of hot material” in the central uplift. This is not the same as a melt chamber. Indeed, impact melting is completed different from endogenic igneous activity. The melting occurs during decompression from high shock pressures. This melting occurs within a zone around the point of impact and that melt is then driven along the base of the transient cavity. After crater formation ceases, the melt ends up as crater-fill deposits (i.e., forming a layer or as patches in breccias) lining the crater interior, in ejecta deposits, and in small dykes intruded into the crater floor. Even in craters as large as 250 km on Earth, there is no evidence whatsoever for melt chambers within which is essentially the central uplift. See this review article for an overview of the processes and products of impact melting: Osinski G. R., Grieve R. A. F., Bleacher J. E., Neish C. D., Pilles E. A. and Tornabene L. L. 2018. Igneous rocks formed by hypervelocity impact. *Journal of Volcanological and Geothermal Research* 353:25– 54.

Central uplift rocks can be hot and any water within them can certainly be above the boiling point, but this does not constitute a melt chamber. This must be revisited.

L215-216: Be very careful with statements like this. Hydrocode models have a hard time modelling where melt and ejecta end up. Much of this is resolution. Indeed, there should be melt-bearing crater-fill deposits within and throughout Occator Crater’s floor.

L264-266: See previous comments. I think the authors have good evidence for the formation of the faculae by impact-generated hydrothermal activity (but see above comment about this) but there is no evidence from over a century of study of craters on Earth that melt chambers at depth exist.

L308: The authors are not very explicit about this, but these lobate materials on their geological map and cross sections are exactly what one would expect to see in a complex crater like this – i.e., crater-fill impact melt rocks and melt-bearing breccias. See the review article cited in a previous comment.

L391: Pits resembling these are common in craters on Mars, and have also been recognised on Vesta and Ceres, where they are interpreted as being formed from volatile release from volatile-rich impact melt-bearing breccias. Such a mechanism should be discussed. E.g., see:

Denevi, B.W., Blewett, D.T., Buczkowski, D.L., Capaccioni, F., Capria, M.T., De Sanctis, M.C., Garry, W.B., Gaskell, R.W., Le Corre, L., Li, J.-Y., Marchi, S., McCoy, T.J., Nathues, A., O’Brien, D.P., Petro, N.E., Pieters, C.M., Preusker, F., Raymond, C.A., Reddy, V., Russell, C.T., Schenk, P., Scully, J.E.C., Sunshine, J.M., Tosi, F., Williams, D.A., Wyrick, D., 2012. Pitted terrain on Vesta and implications for the presence of volatiles. *Science* 338, 246–249.

Hibbard, S. M., Osinski, G. R., and Tornabene, L. L., 2018. Crater-Related Surface Morphologies of the Ikapati Crater, Ceres (abstract #2761). 49th Lunar and Planetary Science Conference. Woodlands, Texas, March 2018.

Tornabene, L.L., Osinski, G.R., McEwen, A.S., Boyce, J.M., Bray, V.J., Caudill, C.M., Grant, J.A., Hamilton, C.W., Mattson, S., Mouginiis-Mark, P.J., 2012. Widespread crater-related pitted

materials on Mars: Further evidence for the role of target volatiles during the impact process. *Icarus* 220, 348–368.

Reviewer #2 (Remarks to the Author):

This is a good and useful paper, and I hope it finds its way to publication— after major revisions. It is straightforward— geologic mapping and photo interpretation, with aid of topography and modeling. I have struggled with it, firstly in that it is a very exciting topic, and there are many fascinating observations presented in the paper, but then there are some key weaknesses as presently developed. The images are spectacular (even if as I say below, I wonder whether there are the highest resolution versions available— if so they would help a lot). If published, it will represent what will likely be the definitive analysis of Dawn’s imaging of the fascinating bright deposits (faculae and also associated dark terrain features) of Ceres’ Occator Crater. I have three basic categories of major comments, as summarized here and then elaborated upon: (1) The figures can be improved in a few ways. (2) The topography we see today would likely have been profoundly affected by dehydration if the whole deposits have dehydrated (rather than just a surface veneer). And (3) there seems to be an ambiguity about plausible interpretations of craters— are they endogenic, exogenic, and which are which? The compelling evidence presented here and also in the companion paper by Schenk et al. for a rich variety of cryovolcanic features would lend some support to an alternative cryovolcanic interpretation of some crater interpreted here to be due to impact. The improvements to the figures are probably easily done. The ambiguities raised by points 2 and 3 run deeper in terms of the analysis and interpretation. At a minimum, the ambiguities and implications of alternative interpretations and processes should be clearly specified.

Comment details:

1. Some figures could be much improved (but easily done):

I am unsure whether the review copy’s figures are the same as what will ultimately be published, so maybe the fix is already in the pipeline (I hope. Otherwise there is a small effort that needs to be made)... But if the higher res figures are already available, I would like to see them; it may qualitatively affect the photointerpretation and my comments!

— Image resolution in the highest res figures appears to be about 6 or even 8 m/pixel (even if they are reported out at an oversampled 3 m/pixel; or if data processing, e.g., mosaic development, has killed some of the original resolution). The text says that 3 m/pixel scenes exist. I’m thinking that especially in the supplement some higher resolution images could be provided if the data exist at higher res. This would help with photo interpretation, for example, impact versus endogenic craters, and fractures versus fracture-controlled eruptive deposits. Are higher resolution images available?

— Figure 1: This map, especially the upper panel, should be made available at higher resolution, maybe in the supplementary materials. Blowing up the PDF to 400% size doesn’t help; I just see a lot

of blurry and illegible details that would be nice to see. A further improvement is possible if the practices followed in terrestrial and most Mars mapping were followed: Give each unit an abbreviated two or three letter name and put a few unobtrusive but legible labels in the map on those various units. The color scheme should be improved. There aren't an extremely large number of units, so we should not be forced into doing our own photoshop color matching to distinguish some of the purple units; and the same for some green units.

2. Dehydration will have affected the topography and maybe contributed to "weathering" and mass wasting, but also could affect some basic process and geochronological interpretations. I need to explain in some detail, so sorry for this long explanation:

These chemical systems— assuming they were aqueous and not anhydrous "molten salts," would have involved large volume losses when the frozen system dehydrates, so that the original topography was not even nearly the same as the present topography— not by a long shot; and furthermore, the dehydration process likely would have driven disaggregation of the salty residue, and this could be a primary cause of the weathering that the authors infer. As the composition is thought to be ammonium chloride and sodium carbonate, and these are thought to be salts left by dehydration of a frozen aqueous brine, it would be worth pointing out that volume of the deposits would have decreased dramatically, and the residual salts would be completely broken up by the dehydration. The binary water-sodium carbonate eutectic (between ice and decahydrate) is near 6.1 weight% sodium carbonate, -2.2 C, and roughly 5 millibars water vapor pressure. The binary system also exhibits a series of three peritectics at increasingly high temperatures and increasingly high water vapor pressures— between the deca- and heptahydrates, the hepta- and monohydrates, and the monohydrate and anhydrous sodium carbonate. All of these peritectics are in the vicinity of 30 weight percent sodium carbonate, 70% water. For example, the 'least aqueous' peritectic in this system is between the monohydrate and anhydrous sodium carbonate, and that solution is 31.3 weight % Na_2CO_3 + 68.7% H_2O , and occurs at +107.5 C and about 1.1 bars vapor pressure. It is not necessary that the brine was at a eutectic or peritectic, but these are likely compositions. All are aqueous, but only the ice-decahydrate eutectic contain ice as a stable phase, so if we are talking about derivation from some large frozen icy ocean or hydrosphere, the ice-sodium carbonate eutectic is what we may be talking about. If this is the case, it should be recognized that dehydration of the frozen eutectic brine— if approximated as the binary water-sodium carbonate system— would involve about 94% mass loss in stages, with a sequence of loss first of ice, then conversion of decahydrate to lower hydrates and finally to anhydrous sodium carbonate; this process, if it goes to completion, would cause something approaching 90% volume loss, and what remains might crumble to powder, or at least would go through a series of disaggregation stages (but maybe there would be annealing at each dehydration stage if a liquid was involved, i.e., without direct dehydration, which can happen if the H_2O vapor pressure is low in the dehydrating deposit). In any event, the topography now seen would not be the topography that first formed. Now, the system is not binary. There's ammonium chloride, too. The binary ammonium chloride-water system lacks hydrates to the best of my knowledge but has two solid anhydrous chloride phases— the alpha and beta phases.

There is a eutectic between water ice and alpha ammonium chloride near 19% NH₄Cl + 81% H₂O at - 15.69 C; and an aqueous peritectic between the alpha and beta phases near 60.5% NH₄Cl + 39.5% H₂O at +184.5 C. I am not sure about the ternary system, but actually it cannot be ternary. There must also be ammonium carbonate, though it has not been reported, and very possibly— as most similar natural systems on Earth also contain— there is likely to be sodium chloride and both ammonium and sodium bicarbonate. Basically, I think the system is apt to be pretty complex, but just going with what has been reported, the actual system will have a series of eutectics and peritectics, a variety of both binary and ternary solids, a lot of hydrate phases, and in all cases the complete dehydration of the solid brines would involve a large volume loss. If dehydration has affected only the surface veneer, then the immense volume losses I mention would only pertain to that veneer. But if the whole deposit dehydration, topographic collapse by an order of magnitude or at least a factor of several is likely. In fact original domes can collapse to holes (the hydrated salt version of glacial kettles). It can be that extreme.

Line 71: "... salts in solution and blocks of unmelted silicates and salts...." It would take very little silicate mass to make mixed salts and silicates tend to sink through brines, in fact very rapidly, no matter how salt-concentrated they are. Salts (even without silicates) also would tend to sink very readily; the exception is ice mixed with salts, such as frozen ice-salt hydrate eutectics, which can float; or brine equivalents of debris flows, where fine grained mass suspended in liquid makes a viscous slurry in which denser stuff can be entrained like rocky boulders in Earth's aqueous debris flows. If the flows are basically like debris flows, the authors can say something like "low albedo lobate deposits on the crater floor may be impact-melt of saturated aqueous saline brine mixed with fine-grained solid silicates and salts, providing a matrix in which boulders of unmelted salts and silicates were suspended, rather like debris flows." Or whatever may please the authors and their interpretation.

Lines 86-88: "Occator's domes and mounds may originate from eruptive and/or frost-heave-like processes derived from the solidification and expansion of the lobate material¹⁸." Expansion only occurs in aqueous saline solutions if they are water-rich and form abundant ice in addition to salts or salt hydrates. So if volume expansion is the cause of these mounds, then when dehydration occurs (if it occurs to deep levels), there would be volume reduction of the whole deposit.

Lines 90-93: "Vinalia Faculae are associated with a prominent set of fractures, from which the faculae-forming brines were proposed to originate^{13,15,20-22}. However, our XM₂-based geologic mapping reveals that these fractures cut through the Vinalia Faculae, suggesting that the fractures postdate, and did not provide conduits for, the faculae-forming brines (Figs. 1 & 4)."

Maybe there is more that could be discerned at 3 m/pixel that I cannot see at the figures' apparent resolution of roughly 6 or 8 m/pixel. But what I see looks a lot like parts of fracture-controlled Lunar Crater Volcanic Field, Nevada. Those fractures are associated with Basin and Range faulting. What could look on Ceres like simple fractures may be some small amount of eruptive material that came

from the fractures. And for what likely are shallow sources in the case of Ceres, eruptive activity could very easily reactivate faults. So I am fine with saying that maybe the fractures post date the deposits, but I do not see anything definitive that rules out fracture-controlled eruptions, with or without reactivation after or during the eruptions. The manuscript does not absolutely rule that out, but I don't think the idea of post-eruptive fracturing should be a strong point— just balance out the ideas, in my view, unless the authors are able to see more in their images that I am not seeing in what was provided.

Lines 164-168: "It is not possible to gain such fine-scale resolution in the model ages derived from crater size frequency distributions: statistical errors of Cerean model ages are sometimes as low as a few hundreds of thousands of years, but they are typically on the order of a few millions of years or more, and do not include the larger, unquantifiable, chronology calibration errors." The accuracy of any crater-count-based geochronologic analysis is completely dependent on the craters being impact craters, not endogenic craters. At the given resolution, I don't see that it is possible to be sure, especially since there has been partial infilling of some crater interiors and exteriors (leaving the rim exposed to define that there is a crater). I attach an example drawn from this paper, and then offer two potential terrestrial volcanic crater analogs. The Lunar Crater Volcanic Field has aligned craters, like some areas in Occator Crater (as the paper points out), and the Pinacate Volcanic Field has some partly infilled crater interiors and aggradation also outside the craters— sand instead of salts. The scales and image resolution that I show are not that different from the Occator Crater examples in the paper. It's hard for me to be certain what is impact and what is endogenic inside Occator Crater. I do believe there are impact craters, but it seems likely that there are endogenic ones, too, and how to isolate the two populations of craters is problematic in my view.

Lines 184-191: The interpretive inferences drawn in this paragraph depend on the observations of the topographic relief seen today being the same or similar to that when the deposits were emplaced. As I noted above, the volume losses of these frozen saline brines may have been enormous, enough to affect the topography significantly. The methodology used to draw the inferences as stated in this paragraph are dependent on the devolatilization not driving large volume changes; for instance, if volatile loss is confined to the upper surface veneer. Pending adequate time dependent thermal models and thermodynamic stability and vapor diffusion models, we won't really know how deep the devolatilization goes; hence, we can't know whether the implicit assumption needed for this paragraph is fulfilled or is implausible. Therefore, it should be explicitly stated somewhere that this inference is dependent on there not having been much volatile loss and differential topographic deflation. (except a surface veneer— fine, it can dehydrate, so long as resulting dehydrated salt powder does not slough off and expose new material to dehydration)

Line 225 and elsewhere: In discussions of mass wasting, consider that dehydration of salts would also likely disaggregate them, especially if it occurred by production of a vapor phase rather than a liquid phase (a liquid would possibly anneal or cement the less hydrated solids). The enormous

volume losses and changes of crystal structure require the breakdown of the solid framework. This is well known from dehydration of hydrated salts: if it occurs under hyperarid conditions, a powder is formed. If liquids form at peritectics, then the dehydrated or less hydrated residue can be re-cemented.

Figure S2. Beautiful image! I am not so sure that the mapped impact craters can be reliably distinguished from endogenic craters. See for instance, Pinacate Volcanic Field, Mexico and Lunar Crater Volcanic Field, Nevada. I attach a PDF showing some examples (from Google Earth). On Ceres, the more distinctly or most plausibly endogenic craters and fractures, or fracture-aligned eruptive deposits, look plausibly like cinder cones (with infill and partial burial), maars, ash and basalt flow deposits controlled by tectonic fractures in these terrestrial volcanic fields. Obviously impact craters are also abundant on Ceres, so the question when dealing with these faculae is: what rimmed or ringed depressions or features are impact, and which are volcanic?

Lines 267-276 (final main text paragraph). This might be a reasonable place to cite some relevant papers:

Regarding geomorphological evidence of cryovolcanism on icy satellites and comets:

Greenberg, R.; Croft, S. K.; Janes, D. M.; Kargel, J. S.; Lebofsky, L. A.; Lunine, J. I.; Marcialis, R. L.; Melosh, H. J.; Ojakangas, G. W.; Strom, R. G., 1991, Miranda, in: Uranus, J.T. Bergstralh, E.D. Miner, and M.S. Matthews (Eds.), U. of Arizona Press, p. 410.

Croft, S.K., J.S. Kargel, R.L. Kirk, J.M. Moore, P.M. Schenk, and R.G. Strom, 1995, The Geology of Triton, in Neptune and Triton, D.P. Cruikshank, ed., pp. 879-947, University of Arizona Press, Tucson.

Lopes, R.M.C., R. L. Kirk, K. L. Mitchell, A. LeGall, J. W. Barnes, A. Hayes, J. Kargel, L. Wye, J. Radebaugh, E. R. Stofan, M. A. Janssen, C. D. Neish, S. D. Wall, C. A. Wood, J. I. Lunine, and M. Malaska, 2013, Cryovolcanism on Titan: New results from Cassini RADAR and VIMS, *Jour. Geophys. Res.: Planets*, 118, 1-20.

Mosis, O., A. Guilbert-LePoutre, B. Brugger, L. Jorda, J.S. Kargel, A. Bouquet, A.-T. Auger, P. Lamy, P. Vernazza, N. Thomas, and H. Sierks, 2015, Pit formation from volatile outgassing on 67P/Churyumov-Gerasimenko, *Astrophys. J. Lett.* 814 (1). <https://doi.org/10.1088/2041-8205/814/1/L5>

Desch, S.J., and M. Neveu, 2017. Differentiation and cryovolcanism on Charon: A view before and after New Horizons, *Icarus* 287 (2017) 175–186.

Regarding predicted occurrence and chemical properties of brine cryovolcanism and cryolavas:

Kargel, J.S., 1991, Brine volcanism and the interior structures of asteroids and icy satellites, *Icarus*, 94, 368-390.

Kargel, J.S., 1992, Ammonia-water volcanism on icy satellites: phase relations at 1 atmosphere, *Icarus*, 100, 556-574.

Porco, C., Helfenstein, P., Thomas, P. C., Ingersoll, A. P., Wisdom, J., West, R., Neukum, G., Denk, T., Wagner, R. 2006. Cassini Observes the Active South Pole of Enceladus. *Science*. 311 (5766): 1393– 1401.

Marion, G., J.S. Kargel, D.C. Catling, and J.I. Lunine, 2012, Modelling ammonia-ammonium aqueous chemistries in the Solar System's icy bodies, *Icarus* 22, 932-946.

The photointerpretation of craters on Ceres would be easier or more reliable if the reader had 3m/pixel images rather than what look more like 6 or 8 m/pixel images. Are these figures, as reproduced, the highest resolution scenes available (and presented at full resolution?) If these images truly are 3 m/pixel, something in the data processing pipeline seems to have degraded the true resolution by roughly a factor of 2 (or worse) even if they are reported out at 3 m/pixel.

Figures 4 and S2. The largest craters occur on fractures or fracture-controlled eruptive deposits. The craters are consistent with the appearance of rimmed maars, like Lunar Crater, Nevada (see attached PDF). This could be noted. Alternatively, they could be impact craters, but I agree with the mapping that they are not included as impact features. My questioning of the endogenic vs exogenic interpretation then extends to smaller craters, also. What is the geometry of solar illumination (azimuth and elevation) in this area?

About the crater distribution in Figure S2: See the attached PDF of Pinacate Volcanic Field. In that case, of course the craters and other volcanic features lie within the volcanic field (they define the volcanic field). So in Figure S2a, why is it that the authors are so certain that the mapped impact craters really are impact craters? I don't think the resolution is there to make a unique and confident determination. So what to do? I think at a minimum the authors need to state that there is a difficulty to make a definitive interpretive disambiguation of impact versus endogenic craters, and the geochronological analysis is dependent on having made a successful identification of the two types of craters.

In Figure S2b (and S1d) I have a similar difficulty to discern clearly whether the linear features are fractures or fracture-controlled eruptive materials. To my eyes, they seem more likely the latter, rather than fractures that purely and simply cut across the deposits. Then the text discussion of the geochronological/stratigraphic sequence is not so clear to me, and the former interpretation that is cited may well be accurate. But it is not a clear situation. Again a true 3 m/pixel would help.

Figure S3. There are four cyan-to-purple or violet color gradations. The cyan unit (bright material fresh) is easily distinguished. The other three purple or violet color units are very difficult to distinguish. Likewise, there are the green to olive colored units; they are very difficult to distinguish.

Reviewer #3 (Remarks to the Author):

Chloe Beddingfield's review for Scully et al. "The Varied Sources of Faculae-Forming Brines in Ceres' Occator Crater, Emplaced via Brine Effusion in a Hydrothermal System"

General Comments:

Overall, I enjoyed reading the manuscript. The manuscript is well written and presents important new information regarding the properties, emplacement mechanisms, and timing relationships of Occator Crater Faculae. The methodology for the manuscript is presented in clear detail and the results and interpretations are important and interesting. I greatly appreciate how much additional information and detail was provided in the supplementary material.

Some more discussion should be included about what would cause Vinalia Faculae, and the brine reservoir, to be so much more brine-limited than the central faculae. You mention that Vinalia Faculae could be sourced from a deep, long-lived brine reservoir that existed prior to the Occator-forming impact. However, it is unclear why this source of the brine would be so much more limited.

Your geologic maps in Figure 1 are very well detailed. Based on looking at the geologic maps, I can clearly see the relationship between the fractures and Vinalia Faculae. Something I'm wondering about is, why are there not more faculae in other locations that are severely fractured. For example, the region to the southwest of Pasola Facula is even more fractured than the region of Vinalia Faculae, and yet faculae are not present here. Why? Is there some implication here for the extent of the pre-existing brine reservoir? What other possibilities are there? Discussion about this is needed.

In the abstract, it would be useful to mention the interpretation that Pasola Facula was not emplaced simultaneously with and was not originally connected to Cerealia Facula, and that Cerealia Tholus has an intrusive origin. Also, you say in the abstract that the faculae were emplaced from numerous localized origins. However, in the manuscript you only discuss two origins, so wouldn't it be better to specify two localized origins than use the word numerous? Also, I didn't get the impression from reading the

manuscript that the pre-existing brine reservoir is considered to be localized, although it is described this way in the abstract. In addition, I think it would be useful to point out in the abstract that you are providing a new geologic map.

Figure 5 is a very nice figure that really helps to visualize your interpretations. However, most of the faults and fractures do not reach the deep brine reservoir in the illustration. If evidence for this fault and fracture depth geometry has been tested, then I suggest mentioning that reference in the caption. It seems that the fractures sourcing the impact induced melt chamber should reach similar depths as the surrounding fractures and faults, especially since you mention in the manuscript that there may be some additional contribution from the deep brine reservoir in this location. Some short explanation is needed here regarding why the depths of these faults vary spatially, or just a reference if one exists. From this figure I see that the illustrated fractures underlying Vinalia Faculae are the only ones that reach the deep brine reservoir, which could be a possible explanation for why faculae are not present in other fractured regions. However, this information is not discussed in the manuscript results or discussion. Additionally, I'm skeptical that this is the best explanation based on the similar plan view lengths of fractures/faults in both the Vinalia Faculae region compared to fractured regions elsewhere where faculae are not present. It seems to me that these fractures/faults likely reach similar depths. This possibility should be mentioned and other possibilities should also be discussed to explain the lack of faculae in these locations. For example, perhaps the deep brine reservoir is not present, or is deeper in the subsurface in this region than elsewhere, etc. One other very minor comment is that one of the normal faults appears listric (third fault from the right), and I suggest changing the geometry of this fault to match its neighbors.

Regarding the data availability, it would be so useful to the community if the digital and georeferenced version of the map provided in Figure 1 was made available for import into a GIS software so others could build off the great work you've done and so that it's preserved long term. Although the detail of the map shown in Figure 1 is fantastic, some of the details are lost due to the restricted size that the figure can be on the page. A GIS version of the map would be very useful for those wanting to see a bit more detail and to use your geologic map to investigate specific features. I recommend providing a georeferenced and GIS importable version of this map.

Minor Comments:

LINE 23: I suggest providing more information on what specific questions remained about the faculae that you address in your manuscript. Perhaps the source, timing relationships, etc? Providing more information here will help the reader immediately see how important your work and results are.

LINE 24-25: This sentence can be removed if you need additional space in the abstract.

LINE 42: Add the word “the” before “largest”

LINE 48: It’s unclear from this sentence what is meant by a ledge deposit. I suggest defining a ledge deposit here.

LINE 49: What is visual normal albedo? Do you mean visible? Although it may be discussed in Reference 9, I suggest rephrasing the sentence here to specify what is meant by visual normal albedo.

LINE 50: I suggest adding the words “composed of” before “sodium carbonate”

LINE 76: “allows” should be “allow”

LINE 105-107: A subject is needed after the word “this”

LINE 119: You state that flow fronts are not visible within any of the faculae, and that this can be explained by multiple overlapping/intermingling flows. However, this argument is not entirely convincing since I can think of examples of overlapping flows that still have clearly visible flow fronts. I think this sentence could use a reference.

LINE 136: The word compressional should be replaced with contractional

LINE 136: It should be mentioned that although contractional linear features are not observable, compressional stresses can be present without the presence of contractional features. Commonly, contractional structures are often not observed since they require larger differential stresses to form than extensional structures. Some good references for this are Gold (1977), Hobbs (1974), and Haynes (1978).

LINE 148: Since the word correlation implies a relationship backed up by regression analysis results, the word correlation here should be replaced with relationship. I have the same comment for Line 149.

LINE 149 – 151: I think this is a great observation. Interesting!

LINE 152 – The sentence here needs to be rephrased. The part about Cerealia Facula ranges in thickness from <3m, through ~5.5m or ~31 m, to >50 m is confusing. What does “through ~5.5m or ~31 m” mean? Although this may be discussed in more detail in Methods 3-4, it needs to be made clear in this sentence.

LINE 154: the word “illustrates” should be “illustrate”

LINE 154-156: The word “Cerealia Tholus” is used twice in this sentence and just sounds a bit redundant. I suggest rephrasing the sentence to fix this issue.

LINE 159-163: Very interesting result and interpretation!

LINE 163: How was 10,000s -100,000s of years derived? More information for how these values were estimated need to be provided here. I realize that in the methodology you refer to some personal communication as a reference. However, this reference should also be included here and some explanation for how these values were derived should also be mentioned.

LINE 175: You mention earlier in the manuscript that the lack of a clear termination may be due to overlapping flow fronts. I think this should also be mentioned and discussed here as a possibility along with the low viscosity possibility. Could multiple higher viscosity brines that overlap each other so that clear termination scarps are not observable? In this case, is an intrusive origin still the most likely explanation for Cerealia Tholus?

LINE 192: This sentence is redundant with a sentence earlier in the manuscript on line 152. I suggest removing one of the sentences. If you keep this sentence, then I have the same comment here as mentioned above for line 152.

LINE 201: A subject is needed after the word “this”.

LINES 209-212: Very fascinating results!

LINE 228: A subject is need after the word “this”. I have the same comment for Lines 251, 450, and 477.

LINE 294: Include a reference for ESRI ArcMap/ArcScene

LINE 298: Your technique using 3D view to more accurately place contacts and stratigraphic relations is neat!

LINE 225: What indicates that the brine reservoir would be deeper than the Cerealia and Pasola-Facula-forming brines. Although the Cerealia and Pasola Faculae are sourced from an impact induced melt chamber, couldn't the Vinalia Faculae brine reservoir also be shallow? Some explanation is needed here and/or a reference.

LINE 303-306: I don't think the last sentence of this paragraph is needed. I suggest removing it.

LINE 310: I suggest replacing "shortly" with the actual timing of when this event took place. By shortly do you mean days? Years? Thousands of years?

LINE 311: The words "comparatively" is used, but it's unclear what comparatively is referring to. Is the lobate material water ice rich compared to the surrounding terrain? Or, to the compositions of the original flows around the crater interior? I suggest rephrasing the sentence to clarify this.

LINE 616: I have one very minor suggestion for Figure 1b. The words "Material" and "here" are not needed. If you'd like to reduce the amount of annotation on your map, you could remove these words in each of the white boxes and include a short explanation in the figure caption.

LINE 622: In Figure 2, elevation values are needed on the contour lines.

LINE 626: The word correlation should be replaced with spatial relationship or something similar since statistical tests were not performed to quantify a correlation.

LINE 628: Although it is mentioned in the figure caption, it would be so much easier to interpret the figure if a legend was provided for the yellow, orange, and white material on Figure 2b.

LINE 650: Figure 4 looks great, but an arrow is needed that points to the landslide in the bottom right of the figure because it's not immediately obvious. Also, this is very minor, but it's hard to see the white box around the vent-like structure. I suggestion changing the box color.

Response to Reviewers for Scully et al. “The Varied Sources of Faculae-Forming Brines in Ceres’ Occator Crater, Emplaced via Brine Effusion in a Hydrothermal System”
(NCOMMS-19-23926)

Reviewers’ comments are in black text in this response to reviewers document.
Author responses are in blue text in this response to reviewers document.

Changes to the manuscript are in blue text in the revised manuscript.

Response to Reviewer #1:

General Comments:

1. This was an interesting and generally well written paper to read. Minor comments are provided in the attached annotated pdf while some more major comments are below. My two most critical comments are highlighted in bold below and relate to the very short presentation of the impact-generated hydrothermal system case and the talk of an impact melt chamber, which is not consistent with what we know about impact craters on Earth and other planetary bodies.

Thank you for taking the time to provide a thorough and helpful review of our manuscript. Below we address the comments about the impact-generated hydrothermal system and the impact melt chamber, along with the other comments.

2. L47: The authors use the term “central pit”. This term has been used for craters on Mars and other bodies with various widely differing explanations for why there is a pit present versus a central peak. To avoid confusion I’d suggest using the term “central depression” as you haven’t really justified as to why this morphological feature is a “pit”.

We understand that the term ‘central depression’ would distinguish this feature from the vast body of literature about central pits on Mars and the icy satellites, but the term ‘central pit’ has already been used extensively in the literature about Occator crater. For example, it is used throughout the already published Icarus special issue about Occator crater (issue #320). Thus, we think that it would be inconstant to change the term now, and leave it as ‘central pit’.

3. L58 and elsewhere: The authors make quite a few very unequivocal statements using “was”. Given everything remains an interpretation or a hypothesis until ground-truth from surface missions happen, I would urge the authors to tone down some of these statements and use phrasing such as “have been interpreted to have formed by X” etc.

We have made these suggested changes on lines 58-61 and lines 72-76 (see minor comments #3 and #4).

4. L78-81: Given the main conclusion of this paper is that Occator Crater’s faculae formed from impact-generated hydrothermal activity, I think this sentence represents a huge oversimplification of the topic. I also think that this is putting the cart before the horse and I would strongly urge the authors to lay out their observations first and then explain why they think impact-generated hydrothermal activity was the process responsible for forming the faculae. With respect to this sentence, the hottest locations in impact craters (by far) are not central uplifts, but rather the crater-fill impact melt-bearing breccias and melt rocks, so at the very least, this needs to be reworded. In the Osinski et al. paper you reference, the various locations within and around craters where hydrothermal deposits are located is presented. I think a short summary of this is key here.

We have edited and rearranged this paragraph as follows, to address this comment: “Instead of identifying one centralized source region for the faculae, the XM2 data allow us to observe numerous localized bright material point features surrounding Cerealia and Vinalia Faculae, which we map as the faint mottled bright material surface feature (Figs. 1 & S1). We also observe that faculae tend to occur within the same general regions of the crater floor as fractures, domes and mounds, and that Cerealia Facula is concentrated within, and surrounding, the central pit (Fig. 3). Occator’s domes and mounds may originate from eruptive and/or frost-heave-like processes derived from the solidification and expansion of the lobate material(15,22). Examination of terrestrial impact-derived hydrothermal deposits shows that hydrothermal deposits mainly occur in crater-fill materials, the inside and outer margin of central uplifts, the ejecta, the crater rim and in crater-lake sediments(23). The distribution of the faint mottled bright material around Occator’s central pit is analogous to the uneven distribution of mounds, which are interpreted to be hydrothermal, around the central structure of the martian crater Toro(25). Moreover, impact-derived fracture networks are found to be key drivers of the location of impact-induced hydrothermal activity(23). A

similar process appears to occur on Ceres: the relationship between Cerealia and Vinalia Faculae and prominent fractures (Figs. 1-4) indicates that pathways to the surface for the faculae-forming brines were likely opened by the prevalent impact-induced fracturing throughout the crater(24). Moreover, excess pressures from partial crystallization of the melt chamber could also initiate and sustain fracturing(15,34).” (lines 88-107)

5. L96-99: As a suggestion, reactivation need not be from some disparate post-impact process. It is known from studies of craters on Earth that faults generated during the few seconds to minutes of the crater forming process, will gravitationally readjust over thousands to millions of years.

Thank you for this suggestion. We have added it to the text: “Moreover, gravitational readjustment of impact-generated faults, as is observed in impact craters such as Charlevoix and Sudbury on Earth(31), could also cause reactivation.” (lines 142-144)

6. P102-103: I am not convinced that this is a vent feature. Do you have topographic data to show that this is comprised of a central depression surrounded by a rim, which would constitute a vent?

Yes, the topography provided in Schenk et al. (this issue, ref. 32) indicates that this structure consists of a low central rise surrounded by linear depressions. Thus, calling it a vent it not the most appropriate terminology, and we have rephrased this section as follows: “There is a candidate centralized source region, or eruptive crater-like structure, in the center of one of the regions of Vinalia Faculae, which possibly sourced the surrounding bright material(16,32) (Fig. 4). This structure consists of a low central rise surrounded by linear depressions, and is discussed in detail in ref. 32.” (lines 147-150)

7. L108-109: Here and elsewhere the authors refer to reference #24. I read this manuscript and I find considerable overlap in terms of the conclusions with this current manuscript. In their abstract the write “Our results suggest that impact heating from the Occator forming impact provides a viable mechanism for the creation of the observed faculae”. I think the authors need to be clearer about how their work differs from this previous/recently published study, given the similar conclusions.

We have edited this section to more explicitly relate our work to previous publications: “Hydrocode simulations predict that the Occator-forming impact would have created a hydrothermal system on water-ice-rich Ceres(26), and previous work found that the

morphology of Cerealia Facula is generally consistent with terrestrial hydrothermal deposits(14). Our aforementioned morphological observations clearly show features that were not well resolved in the pre-XM2 data (e.g. the numerous localized bright material point features), and thus allow us to more definitively confirm the hydrocode modeling predictions(26). Therefore, we find that the faculae are hydrothermal deposits that were emplaced ballistically and as flows, originating from numerous localized brine sources throughout the crater floor (e.g. the bright material point features), rather than from one centralized source region. In addition, some of the localized bright material point features are likely to be splatter deposits from the ballistic emplacement of brines(13-16). We name this process ‘brine effusion’...” (lines 108-119)

8. L206-212: I am having a hard time understanding this text and the Figure 5 and the suggestion of a “melt chamber formed by the impact”. I do not have access to reference #19, but nowhere in reference #31 do I see mention of this and in reference #24, those authors talk about “a ‘plug’ of hot material” in the central uplift. This is not the same as a melt chamber.

Indeed, impact melting is completed different from endogenic igneous activity. The melting occurs during decompression from high shock pressures. This melting occurs within a zone around the point of impact and that melt is then driven along the base of the transient cavity. After crater formation ceases, the melt ends up as crater-fill deposits (i.e., forming a layer or as patches in breccias) lining the crater interior, in ejecta deposits, and in small dykes intruded into the crater floor. Even in craters as large as 250 km on Earth, there is no evidence whatsoever for melt chambers within which is essentially the central uplift. See this review article for an overview of the processes and products of impact melting: Osinski G. R., Grieve R. A. F., Bleacher J. E., Neish C. D., Pilles E. A. and Tornabene L. L. 2018. Igneous rocks formed by hypervelocity impact. *Journal of Volcanological and Geothermal Research* 353:25–54. Central uplift rocks can be hot and any water within them can certainly be above the boiling point, but this does not constitute a melt chamber. This must be revisited.

Citing ref. 31 (Castillo et al., 2019, now ref. 44) here was an unfortunate typo: ref. 33 (Hesse and Castillo-Rogez, 2018, now ref. 42) should have been cited here. Apologies for the confusion. The impact-induced melt chamber that we refer to here corresponds to the red/orange-colored material in Fig. 1 and Fig. 2 of ref. 26 (Bowling et al., 2019), which they

describe as follows: “The hottest material in the crater forms a plug directly beneath the center of Occator...” This material would be above the melting point of water ice, but not above the melting points of the salts and silicates that are also in Ceres’ crust. Hesse and Castillo-Rogez (2018) refer to this material as an ‘impact-induced cryomagma chamber’. The melted material within this chamber would be melted water ice, which, when mixed with the salts in Ceres’ crust, would form a reservoir of brine from which the faculae could be sourced. By using input parameters that are more appropriate for Ceres than those used by Bowling et al. (2019), Castillo-Rogez and Hesse (2018) find that this chamber would cool and solidify in ~12 Myr after the Occator-forming impact. To address this comment, we have corrected the reference (as well as checking/editing all of the references in the manuscript), and have edited this paragraph as follows: “Hydrocode and thermal modelling of the Occator-forming impact predict that impact-melted water ice mixed with salts from Ceres’ crust would form a briny melt chamber in the center of the crater, which would be roughly

20 km in diameter and extend from the shallow subsurface down to ~20 km(24,26,42) (Fig. 5). The central faculae in our geologic map form a roughly 20 km circle, thus fully encompassing this impact-induced melt chamber’s extent. From this consistency between modeling and mapping results, we infer that both Cerealia Facula and Pasola Facula were locally fed by brines sourced in the impact-induced melt chamber.” (lines 249-256)

9. L215-216: Be very careful with statements like this. Hydrocode models have a hard time modelling where melt and ejecta end up. Much of this is resolution. Indeed, there should be melt-bearing crater-fill deposits within and throughout Occator Crater’s floor. The lobate materials are the Cerean equivalent of melt-bearing crater-fill deposits. But here we were referring to the subsurface impact-induced melt chamber. We have rephrased this sentence to make this clear: “Hydrocode modeling indicates that the impact-induced melt chamber is only ~20 km in diameter (24,26,42), thus making it an unlikely source for Vinalia-Faculae-forming brines.” (lines 259-261)
10. L264-266: See previous comments. I think the authors have good evidence for the formation of the faculae by impact-generated hydrothermal activity (but see above comment about this) but there is no evidence from over a century of study of craters on Earth that melt chambers at depth exist.

The response provided to general comment #8, and the associated edits made in the manuscript, address this comment.

11. L308: The authors are not very explicit about this, but these lobate materials on their geological map and cross sections are exactly what one would expect to see in a complex crater like this – i.e., crater-fill impact melt rocks and melt-bearing breccias. See the review article cited in a previous comment.

Agreed. We have edited the text as follows to address this comment: “While the composition of the melted material is different (water ice versus silicate rock), Occator’s lobate material is the Cerean equivalent of crater-fill impact melt rocks and melt-bearing breccias found in the floors of impact-craters throughout the inner solar system(17).” (lines 76-79)

12. L391: Pits resembling these are common in craters on Mars, and have also been recognized on Vesta and Ceres, where they are interpreted as being formed from volatile release from volatile-rich impact melt-bearing breccias. Such a mechanism should be discussed. E.g., see: Denevi, B.W., Blewett, D.T., Buczkowski, D.L., Capaccioni, F., Capria, M.T., De Sanctis, M.C., Garry, W.B., Gaskell, R.W., Le Corre, L., Li, J.-Y., Marchi, S., McCoy, T.J., Nathues, A., O’Brien, D.P., Petro, N.E., Pieters, C.M., Preusker, F., Raymond, C.A., Reddy, V., Russell, C.T., Schenk, P., Scully, J.E.C., Sunshine, J.M., Tosi, F., Williams, D.A., Wyrick, D., 2012. Pitted terrain on Vesta and implications for the presence of volatiles. *Science* 338, 246–249.

Hibbard, S. M., Osinski, G. R., and Tornabene, L. L., 2018. Crater-Related Surface Morphologies of the Ikapati Crater, Ceres (abstract #2761). 49th Lunar and Planetary Science Conference. Woodlands, Texas, March 2018.

Tornabene, L.L., Osinski, G.R., McEwen, A.S., Boyce, J.M., Bray, V.J., Caudill, C.M., Grant, J.A., Hamilton, C.W., Mattson, S., Mouginis-Mark, P.J., 2012. Widespread crater-related pitted materials on Mars: Further evidence for the role of target volatiles during the impact process. *Icarus* 220, 348–368.

We have added the following text to address this comment: “Such endogenic pits could be formed by release of volatiles during cooling of the crater(32), in a manner reminiscent of the formation of pitted terrain on Mars, Vesta and Ceres by degassing of impact-heated volatile-bearing materials(59-62). While the endogenic pits share some morphological similarities

with the pitted terrain (such as a lack of raised rims and irregular shapes), Occator's endogenic pits are more isolated and coalesce less than typical pitted terrain." (lines 471-476)

Minor Comments (from annotated PDF):

1. Line 43: replace the hyphen with "to". Done. (line 43)
2. Line 45: delete the comma. Done. (line 45)
3. Line 58: "was suggested to have been" or similar. "Was" is too unequivocal. We have rephrased this sentence as follows: "Flows are hypothesized to have emplaced the bright material that has a more continuous appearance (corresponding to the continuous bright material geologic unit), while the discontinuous bright material, which is comparatively diffuse, was suggested to have been ballistically emplaced(13-16)." (lines 58-61)
4. Line 70: "has been interpreted as being emplaced". We have rephrased this sentence as follows: "For example, the XM2-based geologic mapping reveals that almost the entire crater interior is coated by lobate material, which has been interpreted to have been emplaced as a slurry of impact-melted water, salts in solution and blocks of unmelted silicates and salts flowed around the crater interior shortly after Occator's formation(13) (Methods 2)." (lines 72-76)
5. Line 78 and onwards: see major comment in accompanying review. Please see the response to general comment #4.
6. Line 162: see above comment about the fact that impact-generated faults can remain active for millions of year after impact, as the crater slowly gravitationally readjusts. We have edited the text as follows to address this comment: "Instead, within Occator's hydrothermal system, the formation of the bright material on the massifs (such as Pasola Facula) can be explained by prevailing hydrologic gradients in the area and/or the transport of hydrothermal fluids along the prevalent fractures formed by the impact and pit collapse(24) (which may have remained open because of gravitational readjustment of impact-generated faults(31))." (lines 198-203)

7. Line 207 and onwards: see my major comment in the attachment. This section needs to be rewritten.

Please see the response to general comment #8.

8. Line 221 and onwards: see other comments – there is no evidence for melt chambers. Please see the response to general comment #8.

Response to Reviewer #2

General Comments:

This is a good and useful paper, and I hope it finds its way to publication— after major revisions. It is straightforward — geologic mapping and photo interpretation, with aid of topography and modeling. I have struggled with it, firstly in that it is a very exciting topic, and there are many fascinating observations presented in the paper, but then there are some key weaknesses as presently developed. The images are spectacular (even if as I say below, I wonder whether there are the highest resolution versions available— if so they would help a lot). If published, it will represent what will likely be the definitive analysis of Dawn’s imaging of the fascinating bright deposits (faculae and also associated dark terrain features) of Ceres’ Occator Crater. I have three basic categories of major comments, as summarized here and then elaborated upon: (1) The figures can be improved in a few ways. (2) The topography we see today would likely have been profoundly affected by dehydration if the whole deposits have dehydrated (rather than just a surface veneer). And (3) there seems to be an ambiguity about plausible interpretations of craters— are they endogenic, exogenic, and which are which? The compelling evidence presented here and also in the companion paper by Schenk et al. for a rich variety of cryovolcanic features would lend some support to an alternative cryovolcanic interpretation of some crater interpreted here to be due to impact. The improvements to the figures are probably easily done. The ambiguities raised by points 2 and 3 run deeper in terms of the analysis and interpretation. At a minimum, the ambiguities and implications of alternative interpretations and processes should be clearly specified.

Thank you for taking the time to provide a thorough and helpful review of our manuscript.

We address points 1, 2 and 3 in the responses to the detailed comments below.

Comment details:

1. Some figures could be much improved (but easily done):

I am unsure whether the review copy’s figures are the same as what will ultimately be published, so maybe the fix is already in the pipeline (I hope. Otherwise there is a small effort that needs to be made)... But if the higher res figures are already available, I would like to see them; it may qualitatively affect the photointerpretation and my comments! Image

resolution in the highest res figures appears to be about 6 or even 8 m/pixel (even if they are reported out at an oversampled 3 m/pixel; or if data processing, e.g., mosaic development, has killed some of the original resolution). The text says that 3 m/pixel scenes exist. I'm thinking that especially in the supplement some higher resolution images could be provided if the data exist at higher res. This would help with photo interpretation, for example, impact versus endogenic craters, and fractures versus fracture-controlled eruptive deposits. Are higher resolution images available?

Yes. For the initial submission, we exported all of the figures at 300 dpi from Adobe Illustrator, and added them into the Word document at that resolution. Unfortunately, the image quality was significantly reduced when combined into the PDF. For this resubmission, we include all of the figures as separate files, which should preserve the original 300 dpi resolution.

2. Figure 1: This map, especially the upper panel, should be made available at higher resolution, maybe in the supplementary materials. Blowing up the PDF to 400% size doesn't help; I just see a lot of blurry and illegible details that would be nice to see.

Agreed. This figure did not turn out well in the combined PDF. Including this figure as one of the separate 300 dpi resolution figures should have fixed this problem. We also have included a highest-resolution JPEG version of the geologic map as a separate supplementary figure (Fig. S14).

3. A further improvement is possible if the practices followed in terrestrial and most Mars mapping were followed: Give each unit an abbreviated two or three letter name and put a few unobtrusive but legible labels in the map on those various units.

We include such abbreviated labels. For example, the continuous bright material is abbreviated to 'bc', which is included at appropriate locations on the geologic map. See Figure 1a.

4. The color scheme should be improved. There aren't an extremely large number of units, so we should not be forced into doing our own photoshop color matching to distinguish some of the purple units; and the same for some green units.

We have made the green and purple colors in the geologic map contrast with one another more in Figure 1, Figure 3, Figure 5, Figure S7, Figure S9 and Figure S12.

5. 2. Dehydration will have affected the topography and maybe contributed to “weathering” and mass wasting, but also could affect some basic process and geochronological interpretations. I need to explain in some detail, so sorry for this long explanation:

These chemical systems— assuming they were aqueous and not anhydrous “molten salts,” would have involved large volume losses when the frozen system dehydrates, so that the original topography was not even nearly the same as the present topography— not by a long shot; and furthermore, the dehydration process likely would have driven disaggregation of the salty residue, and this could be a primary cause of the weathering that the authors infer. As the composition is thought to be ammonium chloride and sodium carbonate, and these are thought to be salts less by dehydration of a frozen aqueous brine, it would be worth pointing out that volume of the deposits would have decreased dramatically, and the residual salts would be completely broken up by the dehydration. The binary water-sodium carbonate eutectic (between ice and decahydrate) is near 6.1 weight% sodium carbonate, -2.2 C, and roughly 5 millibars water vapor pressure. The binary system also exhibits a series of three peritectics at increasingly high temperatures and increasingly high water vapor pressures— between the deca- and heptahydrates, the hepta- and monohydrates, and the monohydrate and anhydrous sodium carbonate. All of these peritectics are in the vicinity of 30 weight percent sodium carbonate, 70% water. For example, the ‘least aqueous’ peritectic in this system is between the monohydrate and anhydrous sodium carbonate, and that solution is 31.3 weight % Na_2CO_3 + 68.7% H_2O , and occurs at +107.5 C and about 1.1 bars vapor pressure. It is not necessary that the brine was at a eutectic or peritectic, but these are likely compositions. All are aqueous, but only the ice-decahydrate eutectic contain ice as a stable phase, so if we are talking about derivation from some large frozen icy ocean or hydrosphere, the ice-sodium carbonate eutectic is what we may be talking about. If this is the case, it should be recognized that dehydration of the frozen eutectic brine— if approximated as the binary water-sodium carbonate system— would involve about 94% mass loss in stages, with a sequence of loss first of ice, then conversion of decahydrate to lower hydrates and finally to anhydrous sodium carbonate; this process, if it goes to completion, would cause something approaching 90% volume loss, and what remains might crumble to powder, or at least would go through a series of disaggregation stages (but maybe there would be annealing at each dehydration stage if a liquid was involved, i.e., without direct dehydration, which can happen if the H_2O

vapor pressure is low in the dehydrating deposit). In any event, the topography now seen would not be the topography that first formed. Now, the system is not binary. There's ammonium chloride, too. The binary ammonium chloride-water system lacks hydrates to the best of my knowledge but has two solid anhydrous chloride phases—the alpha and beta phases. There is a eutectic between water ice and alpha ammonium chloride near 19% NH_4Cl + 81% H_2O at -15.69 C; and an aqueous peritectic between the alpha and beta phases near 60.5% NH_4Cl + 39.5% H_2O at +184.5 C. I am not sure about the ternary system, but actually it cannot be ternary. There must also be ammonium carbonate, though it has not been reported, and very possibly— as most similar natural systems on Earth also contain— there is likely to be sodium chloride and both ammonium and sodium bicarbonate. Basically, I think the system is apt to be pretty complex, but just going with what has been reported, the actual system will have a series of eutectics and peritectics, a variety of both binary and ternary solids, a lot of hydrate phases, and in all cases the complete dehydration of the solid brines would involve a large volume loss. If dehydration has affected only the surface veneer, then the immense volume losses I mention would only pertain to that veneer. But if the whole deposit dehydration, topographic collapse by an order of magnitude or at least a factor of several is likely. In fact original domes can collapse to holes (the hydrated salt version of glacial kettles). It can be that extreme.

Dawn's VIR spectrometer has observed hydrated salts on Ceres' surface (hydrated sodium carbonate, Carrozzo et al., 2018), and hydrated sodium chloride at Cerealia Facula (De Sanctis et al., in review, will be part of this special issue). Hydrated sodium chloride will dehydrate on Ceres' surface in tens of years without another source of hydration, which suggests that there is at least still part of a brine chamber/reservoir present at depth (De Sanctis et al., in review). Thus, it is unlikely that there has been widespread dehydration throughout the whole subsurface, e.g. massive amounts of salt dehydration are unlikely to have formed large-scale topographic features such as the central pit. However, not all salts on Ceres' surface are hydrated, indicating that there has been some dehydration in surficial deposits and/or that not all of the salts were hydrated when they formed. We tried to find a calculation of the depth to which salts would dehydrate on Ceres, but all we could find were references explaining that there will be a stratification in the amount of dehydrated salts with depth, and that hydrated salts will be more stable than water ice at the same depth, and will

stay hydrated in the presence of water ice (Zolotov et al., 2017; Zolotov et al., 2001). However, a calculation for the stability of water ice at depth in Occator crater has been done: Landis et al. (2019) find that the water ice underneath the surface of Occator is covered by a sublimation lag ≤ 1 m thick. Combining this with the aforementioned knowledge of the stability of hydrated salts in comparison to water ice, we can conclude that any hydrated salts that exist at or below the ≤ 1 m thick sublimation lag would stay hydrated. Thus, volume loss due to dehydration would only affect the top ~ 1 m of material, resulting in it not significantly effecting the topography in the crater. We have added the following text to the paper to address this point: “Hydrous sodium chloride has also been observed within Cerealia Facula and, because of its rapid dehydration timescales at Ceres’ surface conditions (tens of years), suggests that at least some brines may still be present in the subsurface(12)... The now-solidified lobate material is comparatively rich in water ice when compared to the surrounding terrain(18), and is covered by a desiccated sublimation lag most likely ≤ 1 m thick(19). Hydrated salts are more stable than water ice at the same depth, and would stay hydrated in the presence of water ice(20,21). Thus, any hydrated salts that exist at or below the ≤ 1 m thick sublimation lag would stay hydrated, meaning that volume loss due to dehydration would not significantly affect the topography within Occator crater.” (lines 52-55 and lines 79-85)

6. Line 71: “... salts in solution and blocks of unmelted silicates and salts...” It would take very little silicate mass to make mixed salts and silicates tend to sink through brines, in fact very rapidly, no matter how salt-concentrated they are. Salts (even without silicates) also would tend to sink very readily; the exception is ice mixed with salts, such as frozen ice-salt hydrate eutectics, which can float; or brine equivalents of debris flows, where fine grained mass suspended in liquid makes a viscous slurry in which denser stuff can be entrained like rocky boulders in Earth’s aqueous debris flows. If the flows are basically like debris flows, the authors can say something like “low albedo lobate deposits on the crater floor may be impact-melt of saturated aqueous saline brine mixed with fine-grained solid silicates and salts, providing a matrix in which boulders of unmelted salts and silicates were suspended, rather like debris flows.” Or whatever may please the authors and their interpretation.

We have added the following text to the section #2 of the methods, which discusses the lobate material in detail, to address this comment: “The slurry may have suspended the blocks of unmelted silicates and salts, in a similar process to a debris flow.” (lines 386-387)

7. Lines 86-88: “Occator’s domes and mounds may originate from eruptive and/or frost-heave-like processes derived from the solidification and expansion of the lobate material¹⁸.” Expansion only occurs in aqueous saline solutions if they are water-rich and form abundant ice in addition to salts or salt hydrates. So if volume expansion is the cause of these mounds, then when dehydration occurs (if it occurs to deep levels), there would be volume reduction of the whole deposit.

The lobate material is rich in water ice in comparison to the surrounding terrain (Prettyman et al., 2019), meaning that the formation of topographic features due to expansion is likely (Schmidt et al., in review). Also, as discussed in response to point #5, dehydration of salts would occur in the top ~1 m of the subsurface, meaning that dehydration would not significantly effect on the topography within Occator crater.

8. Lines 90-93: “Vinalia Faculae are associated with a prominent set of fractures, from which the faculae-forming brines were proposed to originate^{13,15,20-22}. However, our XM2-based geologic mapping reveals that these fractures cut through the Vinalia Faculae, suggesting that the fractures postdate, and did not provide conduits for, the faculae-forming brines (Figs. 1 & 4).” Maybe there is more that could be discerned at 3 m/pixel that I cannot see at the figures’ apparent resolution of roughly 6 or 8 m/pixel. But what I see looks a lot like parts of fracture-controlled Lunar Crater Volcanic Field, Nevada. Those fractures are associated with Basin and Range faulting. What could look on Ceres like simple fractures may be some small amount of eruptive material that came from the fractures. And for what likely are shallow sources in the case of Ceres, eruptive activity could very easily reactivate faults. So I am fine with saying that maybe the fractures post date the deposits, but I do not see anything definitive that rules out fracture-controlled eruptions, with or without reactivation after or during the eruptions. The manuscript does not absolutely rule that out, but I don’t think the idea of post-eruptive fracturing should be a strong point— just balance out the ideas, in my view, unless the authors are able to see more in their images that I am not seeing in what was provided.

Including this figure as one of the separate 300 dpi resolution figures should now allow the morphology of the fractures/pit chains and Vinalia Faculae to be seen in detail. We have updated the text in the paper to address this comment: “We observe that the Vinalia Faculae fractures often broaden into pit chains coated by dark talus, and there is no clear evidence that the Vinalia Faculae bright material originated from the fractures. In contrast, we observe that landslides of bright material, originating from bright outcrops at the pits’ rims, cascade down into the dark pit chains. Disaggregation of dehydrated salts from the top ~1 m of the subsurface could form some of the loose bright material that, after becoming unstable, mass wasted into the pit chains. These observations suggest that the fractures postdate, and did not provide conduits for, the faculae-forming brines. Nevertheless, the fractures that we currently see at the surface could predate the faculae, and have formed conduits for the faculae-forming brines, if they were reactivated following faculae formation.” (lines 128-138)

9. Lines 164-168: “It is not possible to gain such fine-scale resolution in the model ages derived from crater size frequency distributions: statistical errors of Cerean model ages are sometimes as low as a few hundreds of thousands of years, but they are typically on the order of a few millions of years or more, and do not include the larger, unquantifiable, chronology calibration errors.” The accuracy of any crater-count-based geochronologic analysis is completely dependent on the craters being impact craters, not endogenic craters. At the given resolution, I don’t see that it is possible to be sure, especially since there has been partial infilling of some crater interiors and exteriors (leaving the rim exposed to define that there is a crater). I attach an example drawn from this paper, and then offer two potential terrestrial volcanic crater analogs. The Lunar Crater Volcanic Field has aligned craters, like some areas in Occator Crater (as the paper points out), and the Pinacate Volcanic Field has some partly infilled crater interiors and aggradation also outside the craters— sand instead of salts. The scales and image resolution that I show are not that different from the Occator Crater examples in the paper. It’s hard for me to be certain what is impact and what is endogenic inside Occator Crater. I do believe there are impact craters, but it seems likely that there are endogenic ones, too, and how to isolate the two populations of craters is problematic in my view.

We agree, distinguishing between impact craters and endogenic pits within Occator crater is a complex task. We discuss the identification of endogenic pits in part 3 of the Methods

section of the paper: “There are crater-like features in Cerealia Facula that we map as pits (Fig. S11a). They do not have regular bowl shapes and may be vents through which brines were ballistically emplaced, because ballistic eruptions can be easily driven by less than 1% volatiles in Ceres’ low gravity environment(15-16). Such endogenic pits could be formed by release of volatiles during cooling of the crater(32), in a manner reminiscent of the formation of pitted terrain on Mars, Vesta and Ceres by degassing of impact-heated volatile-bearing materials(59-62). While the endogenic pits share some morphological similarities with the pitted terrain (such as a lack of raised rims and irregular shapes), Occator’s endogenic pits are more isolated and coalesce less than typical pitted terrain. To ensure we used impact craters for our thickness estimates and not other depressions, such as these endogenic pits, we only used features with regular bowl shapes, ejecta and raised rims. We use all three criteria to identify impact craters, in order to lower the possibility of false detections. For example, endogenic pits could be surrounded by an ejecta-like deposit, but are less likely to have regular bowl shapes.” (lines 468-480) The model ages derived for various features in Occator that we cite were derived via careful identification of impact craters from endogenic pits by Nathues et al. (2017), Nathues et al. (2019), Nathues et al. (this issue) and Neesemann et al. (2019). However, the difficulty in definitively identifying all impact craters from endogenic pits will likely contribute some additional unquantifiable errors in the derived model ages.

10. Lines 184-191: The interpretive inferences drawn in this paragraph depend on the observations of the topographic relief seen today being the same or similar to that when the deposits were emplaced. As I noted above, the volume losses of these frozen saline brines may have been enormous, enough to affect the topography significantly. The methodology used to draw the inferences as stated in this paragraph are dependent on the devolatilization not driving large volume changes; for instance, if volatile loss is confined to the upper surface veneer. Pending adequate time dependent thermal models and thermodynamic stability and vapor diffusion models, we won’t really know how deep the devolatilization goes; hence, we can’t know whether the implicit assumption needed for this paragraph is fulfilled or is implausible. Therefore, it should be explicitly stated somewhere that this inference is dependent on there not having been much volatile loss and differential topographic deflation. (except a surface veneer— fine, it can dehydrate, so long as resulting dehydrated salt powder does not slough off and expose new material to dehydration)

Please see the response to comment #5: we find that volume loss due to dehydration would only affect the top ~1 m of material, resulting in it not significantly effecting the topography in the crater. We have added the text to the paper to address this point on lines 52-55 and lines 79-85, which occur before the discussion on lines 184-191. (now lines 225-232)

11. Line 225 and elsewhere: In discussions of mass wasting, consider that dehydration of salts would also likely disaggregate them, especially if it occurred by production of a vapor phase rather than a liquid phase (a liquid would possible anneal or cement the less hydrated solids). The enormous volume losses and changes of crystal structure require the breakdown of the solid framework. This is well known from dehydration of hydrated salts: if it occurs under hyperarid conditions, a powder is formed. If liquids form at peritectics, then the dehydrated or less hydrated residue can be re-cemented.

We agree, dehydration of the top ~1 m of the hydrated salts could form some loose bright material. We have added the following text into the manuscript to address this comment:

“Disaggregation of dehydrated salts from the top ~1 m of the subsurface could form some of the loose bright material that, after becoming unstable, mass wasted into the pit chains.”

(lines 132-134)

12. Figure S2. Beautiful image! I am not so sure that the mapped impact craters can be reliably distinguished from endogenic craters. See for instance, Pinacate Volcanic Field, Mexico and Lunar Crater Volcanic Field, Nevada. I attach a PDF showing some examples (from Google Earth). On Ceres, the more distinctly or most plausibly endogenic craters and fractures, or fracture-aligned eruptive deposits, look plausibly like cinder cones (with infill and partial burial), maars, ash and basalt flow deposits controlled by tectonic fractures in these terrestrial volcanic fields. Obviously impact craters are also abundant on Ceres, so the question when dealing with these faculae is: what rimmed or ringed depressions or features are impact, and which are volcanic?

Thank you. Please see the response to comment #9 for a discussion of how we distinguished between impact craters and endogenic pits within Occator crater.

13. Lines 267-276 (final main text paragraph). This might be a reasonable place to cite some relevant papers
 - a. Regarding geomorphological evidence of cryovolcanism on icy satellites and comets: Greenberg, R.; Cros, S. K.; Janes, D. M.; Kargel, J. S.; Lebofsky, L. A.; Lunine, J. I.;

Marcialis, R. L.; Melosh, H. J.; Ojakangas, G. W.; Strom, R. G., 1991, Miranda, in: Uranus, J.T. Bergstrahl, E.D. Miner, and M.S. MaZhevs (Eds.), U. of Arizona Press, p. 410.

- b. Croft, S.K., J.S. Kargel, R.L. Kirk, J.M. Moore, P.M. Schenk, and R.G. Strom, 1995, The Geology of Triton, in Neptune and Triton, D.P. Cruikshank, ed., pp. 879-947, University of Arizona Press, Tucson.
- c. Lopes, R.M.C., R. L. Kirk, K. L. Mitchell, A. LeGall, J. W. Barnes, A. Hayes, J. Kargel, L. Wye, J. Radebaugh, E. R. Stofan, M. A. Janssen, C. D. Neish, S. D. Wall, C. A. Wood, J. I. Lunine, and M. Malaska, 2013, Cryovolcanism on Titan: New results from Cassini RADAR and VIMS, *Jour. Geophys. Res.: Planets*, 118, 1-20.
- d. Mousis, O., A. Guilbert-LePoutre, B. Brugger, L. Jorda, J.S. Kargel, A. Bouquet, A.-T. Auger, P. Lamy, P. Vernazza, N. Thomas, and H. Sierks, 2015, Pit formation from volatile outgassing on 67P/Churyumov-Gerasimenko, *Astrophys. J. Lett.* 814 (1).
[hZps://doi.org/10.1088/2041-8205/814/1/L5](https://doi.org/10.1088/2041-8205/814/1/L5)
- e. Desch, S.J., and M. Neveu, 2017. Differentiation and cryovolcanism on Charon: A view before and after New Horizons, *Icarus* 287 (2017) 175–186.
- f. Regarding predicted occurrence and chemical properties of brine cryovolcanism and cryolavas: Kargel, J.S., 1991, Brine volcanism and the interior structures of asteroids and icy satellites, *Icarus*, 94, 368-390.
- g. Kargel, J.S., 1992, Ammonia-water volcanism on icy satellites: phase relations at 1 atmosphere, *Icarus*, 100, 556-574.
- h. Porco, C., Helfenstein, P., Thomas, P. C., Ingersoll, A. P., Wisdom, J., West, R., Neukum, G., Denk, T., Wagner, R. 2006. Cassini Observes the Active South Pole of Enceladus. *Science*. 311 (5766): 1393–1401.
- i. Marion, G., J.S. Kargel, D.C. Catling, and J.I. Lunine, 2012, Modelling ammonia-ammonium aqueous chemistries in the Solar System's icy bodies, *Icarus* 22, 932-946.

We have added a representative selection of these references: “Cryovolcanism on the icy satellites of the outer solar system (discussed by, for example, refs. 47-50) can be formed by excess pressures from crystallization of reservoirs(34). It is also possible that similar processes to those observed at Occator could occur on the icy satellites...” (lines 344-347)

14. The photointerpretation of craters on Ceres would be easier or more reliable if the reader had 3m/pixel images rather than what look more like 6 or 8 m/pixel images. Are these figures, as reproduced, the highest resolution scenes available (and presented at full resolution?) If these images truly are 3 m/pixel, something in the data processing pipeline seems to have degraded the true resolution by roughly a factor of 2 (or worse) even if they are reported out at 3 m/pixel.

Please see response to comment #1. For this resubmission, we include all of the figures as separate files, which should preserve the original 300 dpi resolution.

15. Figures 4 and S2. The largest craters occur on fractures or fracture-controlled eruptive deposits. The craters are consistent with the appearance of rimmed maars, like Lunar Crater, Nevada (see attached PDF). This could be noted. Alternatively, they could be impact craters, but I agree with the mapping that they are not included as impact features. My questioning of the endogenic vs exogenic interpretation then extends to smaller craters, also. What is the geometry of solar illumination (azimuth and elevation) in this area?

Please see the response to comment #9 for a discussion of how we distinguished between impact craters and endogenic pits within Occator crater. We agree that there are morphological similarities between some of the craters seen in Occator and terrestrial eruptive craters. However, we are hesitant to use the phrase ‘maar’ in our manuscript, because of the specific associations of magma and groundwater/surface water that the word might invoke in the reader. However, we have added the phrase ‘eruptive crater-like’ to the text, to address this comment: “There is a candidate centralized source region, or eruptive crater-like structure, in the center of one of the regions of Vinalia Faculae, which possibly sourced the surrounding bright material(16,32) (Fig. 4).” (lines 147-149) The images of this area have phase angles of $\sim 35^\circ$, and the solar illumination is from the SSE.

16. About the crater distribution in Figure S2: See the attached PDF of Pinacate Volcanic Field. In that case, of course the craters and other volcanic features lie within the volcanic field (they define the volcanic field). So in Figure S2a, why is it that the authors are so certain that the mapped impact craters really are impact craters? I don’t think the resolution is there to make a unique and confident determination. So what to do? I think at a minimum the authors need to state that there is a difficulty to make a definitive interpretive disambiguation of

impact versus endogenic craters, and the geochronological analysis is dependent on having made a successful identification of the two types of craters.

See response to comment #9: we discuss the ambiguities in identification of endogenic pits versus impact craters in part 3 of the Methods section of the paper.

17. In Figure S2b (and S1d) I have a similar difficulty to discern clearly whether the linear features are fractures or fracture-controlled eruptive materials. To my eyes, they seem more likely the latter, rather than fractures that purely and simply cut across the deposits. Then the text discussion of the geochronological/stratigraphic sequence is not so clear to me, and the former interpretation that is cited may well be accurate. But it is not a clear situation. Again a true 3 m/pixel would help.

Including the figures as separate 300 dpi resolution files should now allow the cross-cutting relationship between the fractures/pit chains and the Vinalia Faculae to be clearly observed. Also, see response to comment #8.

18. Figure S3. There are four cyan-to-purple or violet color gradations. The cyan unit (bright material fresh) is easily distinguished. The other three purple or violet color units are very difficult to distinguish. Likewise, there are the green to olive colored units; they are very difficult to distinguish.

We have made the green and purple colors in the geologic map contrast with one another more in Figure 1, Figure 3, Figure 5, Figure S7, Figure S9 and Figure S12.

Response to Reviewer #3: Chloe Beddingfield

General Comments:

1. Overall, I enjoyed reading the manuscript. The manuscript is well written and presents important new information regarding the properties, emplacement mechanisms, and timing relationships of Occator Crater Faculae. The methodology for the manuscript is presented in clear detail and the results and interpretations are important and interesting. I greatly appreciate how much additional information and detail was provided in the supplementary material.

Thank you for taking the time to provide a thorough and helpful review of our manuscript.

2. Some more discussion should be included about what would cause Vinalia Faculae, and the brine reservoir, to be so much more brine-limited than the central faculae. You mention that Vinalia Faculae could be sourced from a deep, long-lived brine reservoir that existed prior to the Occator-forming impact. However, it is unclear why this source of the brine would be so much more limited.

We have edited the text as follows, to address this comment: “Instead, Vinalia Faculae could be sourced from a deep, long-lived brine reservoir, which has been suggested to be present at the base of the crust on the basis of topographic analyses(43) and is supported by thermal modeling(44)... The central faculae would be fed from the impact-induced melt chamber, which is predicted to extend to much closer to the surface than the deep brine reservoir(15,24,26,42,44) (Fig. 5). In contrast, the Vinalia-Faculae-forming brines, sourced in the deep brine reservoir via fractures, would take a longer path to the surface, and thus encounter more potential obstacles to reaching the surface, than the Cerealia- and Pasola-Facula-forming brines. Consequently, the deeper source of Vinalia Faculae is consistent with Vinalia being more brine limited than the central faculae, as indicated by the relatively small thicknesses and volumes of the Vinalia Faculae in comparison to the central faculae (Fig. 1b).” (lines 261-263 and 269-277)

3. Your geologic maps in Figure 1 are very well detailed. Based on looking at the geologic maps, I can clearly see the relationship between the fractures and Vinalia Faculae. Something I’m wondering about is, why are there not more faculae in other locations that are severely fractured. For example, the region to the southwest of Pasola Facula is even more fractured

than the region of Vinalia Faculae, and yet faculae are not present here. Why? Is there some implication here for the extent of the pre-existing brine reservoir? What other possibilities are there? Discussion about this is needed.

We have added the following explanation: “It is likely that Cerealia Facula and Pasola Facula formed in the center of the crater because the impact-induced melt chamber (with likely long-term contributions from the deep brine reservoir) provided a shallow, readily available brine source. It is more difficult to explain why Vinalia Faculae formed in the eastern crater floor and no other faculae formed elsewhere in the crater floor. While there is a relationship between the occurrence of faculae and fractures in Occator’s floor (Fig. 3), faculae are not associated with all of the prominent fractures. Most notably, there are no bright deposits like Vinalia Faculae associated with the cluster of concentric fractures in the southwestern part of Occator’s floor. This cluster of concentric fractures occurs at the boundary between the smooth lobate material and the terrace material with thin lobate mantling (Fig. 1a). Perhaps the terrace material in this region provided a more competent barrier (in comparison to the lobate materials) through which the faculae-forming brines could not flow. Alternatively, perhaps the fractures in this region were configured in a manner that did not provide a viable pathway to the surface. While our current data and models do not provide a definitive explanation for why all of the prominent fractures in Occator do not source faculae, this observation is consistent with our earlier interpretation that the system was brine limited.” (lines 308-323)

4. In the abstract, it would be useful to mention the interpretation that Pasola Facula was not emplaced simultaneously with and was not originally connected to Cerealia Facula, and that Cerealia Tholus has an intrusive origin.

We have added the point about Cerealia/Pasola Facula into the abstract, but decided to not discuss Cerealia Tholus, because the intrusive interpretation is less concrete. The relevant part of the abstract now reads: “The thick central faculae, called Cerealia Facula and Pasola Facula, postdate the central pit, and were not originally connected nor simultaneously emplaced. The central faculae were primarily sourced from an impact-induced melt chamber, with some contribution from a deeper, pre-existing brine reservoir.” (lines 27-31)

5. Also, you say in the abstract that the faculae were emplaced from numerous localized origins. However, in the manuscript you only discuss two origins, so wouldn’t it be better to specify

two localized origins than use the word numerous? Also, I didn't get the impression from reading the manuscript that the pre-existing brine reservoir is considered to be localized, although it is described this way in the abstract.

This part of the abstract was referring to the numerous localized bright material point features that surround the main faculae (lines 88-91). However, we realize that discussion of these localized origination points can easily get confused with the impact-induced melt chamber/deep-brine reservoir origins. Thus, we have removed this segment from the abstract. The sentence now reads: "Here we show that 'brine effusion' emplaced the faculae in an impact-induced hydrothermal system." (lines 25-27)

6. In addition, I think it would be useful to point out in the abstract that you are providing a new geologic map.

Our abstract is already quite long, so we decided to not include this point, because it is discussed shortly after, in the 'Introduction'.

7. Figure 5 is a very nice figure that really helps to visualize your interpretations. However, most of the faults and fractures do not reach the deep brine reservoir in the illustration. If evidence for this fault and fracture depth geometry has been tested, then I suggest mentioning that reference in the caption. It seems that the fractures sourcing the impact induced melt chamber should reach similar depths as the surrounding fractures and faults, especially since you mention in the manuscript that there may be some additional contribution from the deep brine reservoir in this location. Some short explanation is needed here regarding why the depths of these faults vary spatially, or just a reference if one exists. From this figure I see that the illustrated fractures underlying Vinalia Faculae are the only ones that reach the deep brine reservoir, which could be a possible explanation for why faculae are not present in other fractured regions. However, this information is not discussed in the manuscript results or discussion. Additionally, I'm skeptical that this is the best explanation based on the similar plan view lengths of fractures/faults in both the Vinalia Faculae region compared to fractured regions elsewhere where faculae are not present. It seems to me that these fractures/faults likely reach similar depths. This possibility should be mentioned and other possibilities should also be discussed to explain the lack of faculae in these locations. For example, perhaps the deep brine reservoir is not present, or is deeper in the subsurface in this region than elsewhere, etc.

We have added the following text into the manuscript, to address this topic: “It is likely that Cerealia Facula and Pasola Facula formed in the center of the crater because the impact-induced melt chamber (with likely long-term contributions from the deep brine reservoir) provided a shallow, readily available brine source. It is more difficult to explain why Vinalia Faculae formed in the eastern crater floor and no other faculae formed elsewhere in the crater floor. While there is a relationship between the occurrence of faculae and fractures in Occator’s floor (Fig. 3), faculae are not associated with all of the prominent fractures. Most notably, there are no bright deposits like Vinalia Faculae associated with the cluster of concentric fractures in the southwestern part of Occator’s floor. This cluster of concentric fractures occurs at the boundary between the smooth lobate material and the terrace material with thin lobate mantling (Fig. 1a). Perhaps the terrace material in this region provided a more competent barrier (in comparison to the lobate materials) through which the faculae-forming brines could not flow. Alternatively, perhaps the fractures in this region were configured in a manner that did not provide a viable pathway to the surface. While our current data and models do not provide a definitive explanation for why all of the prominent fractures in Occator do not source faculae, this observation is consistent with our earlier interpretation that the system was brine limited.” (lines 308-323) We have also adjusted the lengths of the faults and the depth of the deep brine reservoir, in Figure 5, to address this topic.

8. One other very minor comment is that one of the normal faults appears listric (third fault from the right), and I suggest changing the geometry of this fault to match its neighbors. We fixed the geometry of this fault in Figure 5.
9. Regarding the data availability, it would be so useful to the community if the digital and georeferenced version of the map provided in Figure 1 was made available for import into a GIS software so others could build off the great work you’ve done and so that it’s preserved long term. Although the detail of the map shown in Figure 1 is fantastic, some of the details are lost due to the restricted size that the figure can be on the page. A GIS version of the map would be very useful for those wanting to see a bit more detail and to use your geologic map to investigate specific features. I recommend providing a georeferenced and GIS importable version of this map.

We are currently proposing to create a USGS SIM of Occator crater. The SIM version of the map would contain broadly the same geologic units, linear features etc. as the geologic map in this paper, but would be a more refined product than was possible to make during the Dawn mission. For example, linework would be tidied to ensure that there was only one line per contact, and we would ensure that different regions were mapped at the same level of detail (currently, the faculae regions are mapped in greater detail than other regions of the crater). While we agree that it is important for the community to have access to the GIS files of published geologic maps, we do not want to cause confusion by making the GIS files of this version of the map available shortly before the USGS SIM version, which will be a superior product. We agree that it is important to include a higher quality version of the geologic map than can be contained in the main body of the paper, so we have included a high-resolution JPEG version of the geologic map as a separate supplementary figure (Fig. S14). Accordingly, we have added the following text to the caption of Figure 1: “A high-resolution JPEG, stand-alone version of the geologic map is available as Fig. S14.” (lines 728-729).

Minor Comments:

1. LINE 23: I suggest providing more information on what specific questions remained about the faculae that you address in your manuscript. Perhaps the source, timing relationships, etc? Providing more information here will help the reader immediately see how important your work and results are.

We have edited the text as follows: “Prior to the acquisition of the highest-resolution data of Ceres, questions remained about the emplacement mechanism and source of Occator crater’s bright faculae.” (lines 24-25)

2. LINE 24-25: This sentence can be removed if you need additional space in the abstract. We did need additional space, so have removed this sentence as suggested.
3. LINE 42: Add the word “the” before “largest”. Done. (line 42)
4. LINE 48: It’s unclear from this sentence what is meant by a ledge deposit. I suggest defining a ledge deposit here.

We have edited the text as follows: “Pasola Facula is a bright deposit located on a ledge above the central pit ...” (line 48)

5. LINE 49: What is visual normal albedo? Do you mean visible? Although it may be discussed in Reference 9, I suggest rephrasing the sentence here to specify what is meant by visual normal albedo.

The definition of ‘visual normal albedo’ is somewhat involved, and is not the main point of this sentence. The main point is that the faculae are quantifiably very bright, so we have rephrased this sentence as follows: “The faculae are up to 6 times brighter than Ceres’ average material, as defined by ref. 9.” (lines 49-50)

6. LINE 50: I suggest adding the words “composed of” before “sodium carbonate”.
Done. (line 50)

7. LINE 76: “allows” should be
“allow”. Done. (line 89)

8. LINE 105-107: A subject is needed after the word “this”.

We have rephrased this sentence as follows: “The idea that fractures could control the shape of the candidate centralized source region is analogous to the hypothesis that the shape of polygonal craters on Ceres can be attributed to subsurface fracturing(33).” (lines 152-154)

9. LINE 119: You state that flow fronts are not visible within any of the faculae, and that this can be explained by multiple overlapping/intermingling flows. However, this argument is not entirely convincing since I can think of examples of overlapping flows that still have clearly visible flow fronts. I think this sentence could use a reference.

We agree. This was an initial hypothesis that should have been deleted in favor of the other hypotheses discussed in this paragraph. We have rephrased the beginning of this paragraph as follows: “In the XM2 data, there are no flow fronts clearly visible within any of the faculae, which can be explained by the buildup of ballistic deposits (in the discontinuous bright material) and by the bright material being of a sufficiently low viscosity to form a gradually sloping surface instead of a clear flow front (in the continuous bright material)(34) (Figs. 1a, 2a, S1 & S3).” (lines 155-158)

10. LINE 136: The word compressional should be replaced with contractional. Done. (line 171)

11. LINE 136: It should be mentioned that although contractional linear features are not observable, compressional stresses can be present without the presence of contractional features. Commonly, contractional structures are often not observed since they require larger differential stresses to form than extensional structures. Some good references for this are Gold (1977), Hobbs (1974), and Haynes (1978).

We have added the following explanation: “However, we note that compressional stresses can occur without corresponding contractional features, and that contractional structures are often not present because they require larger differential stresses to form than extensional structures(37-39).” (lines 172-175) Unfortunately I cannot find the references mentioned above. Could you please provide the titles of the references so they can be added to the paper? Thank you.

12. LINE 148: Since the word correlation implies a relationship backed up by regression analysis results, the word correlation here should be replaced with relationship. I have the same comment for Line 149.

Done. (line 187)

13. LINE 149 – 151: I think this is a great observation.
Interesting! Thank you.

14. LINE 152 – The sentence here needs to be rephrased. The part about Cerealia Facula ranges in thickness from <3m, through ~5.5m or ~31 m, to >50 m is confusing. What does “through ~5.5m or ~31 m” mean? Although this may be discussed in more detail in Methods 3-4, it needs to be made clear in this sentence.

We have rephrased this sentence as follows: “...Cerealia Facula ranges in thickness from <3 m at the southern side, to ~5.5 m or ~31 m at the northern side, to ≥ 50 m on top of Cerealia Tholus ...” (lines 191-192)

15. LINE 154: the word “illustrates” should be
“illustrate”. Done. (line 193)

16. LINE 154-156: The word “Cerealia Tholus” is used twice in this sentence and just sounds a bit redundant. I suggest rephrasing the sentence to fix this issue.

We have rephrased this sentence as follows: “The XM2 data illustrate that there is no dark material at the base of the ~50-100 m deep(32) radiating fractures on top of Cerealia Tholus

(Fig. 2a), indicating that the continuous bright material on the uppermost parts of the tholus is ≥ 50 m thick (Fig. 1b).” (lines 192-195)

17. LINE 159-163: Very interesting result and interpretation!

Thank you.

18. LINE 163: How was 10,000s -100,000s of years derived? More information for how these values were estimated need to be provided here. I realize that in the methodology you refer to some personal communication as a reference. However, this reference should also be included here and some explanation for how these values were derived should also be mentioned.

This sentence was intended to mean that there could be a relatively large period of time between emplacement of different areas of faculae, but we understand that the use of specific numbers made this confusing. We have rephrased this section as follows: “...can be explained by prevailing hydrologic gradients in the area and/or the transport of hydrothermal fluids along the prevalent fractures formed by the impact and pit collapse(24) (which may have remained open because of gravitational readjustment of impact-generated faults(31)). It is not possible to gain fine-scale resolution in the model ages derived for the faculae from crater size frequency distributions: statistical errors of Cerean model ages are sometimes as low as a few hundreds of thousands of years, but they are typically on the order of a few millions of years or more, and do not include the larger, unquantifiable, chronology calibration errors(40). Thus, it is plausible that there could be at least a few hundreds of thousands of years separating the emplacement of different parts of the faculae.

Consequently, we interpret that the similarities in reflectance and age between Pasola Facula and Cerealia Facula(36) are because it is material with the same composition(10-12) that was emplaced from multiple sources in the same region over a similar, but not necessarily simultaneous, period of time.” (lines 200-212)

19. LINE 175: You mention earlier in the manuscript that the lack of a clear termination may be due to overlapping flow fronts. I think this should also be mentioned and discussed here as a possibility along with the low viscosity possibility. Could multiple higher viscosity brines that overlap each other so that clear termination scarps are not observable? In this case, is an intrusive origin still the most likely explanation for Cerealia Tholus?

Here we are talking about a significant termination scarp that would be formed around the base of a structure like an extrusive volcanic dome, rather than the smaller, scale flow fronts formed by individual brine flows. We have edited the text as follows for clarification:

“Ahuna Mons, Ceres’ solitary mountain that is interpreted to be an extrusive volcanic dome, is surrounded by a clear termination scarp at its base(41). In contrast, there is only a subtle basal scarp around part of Cerealia Tholus’ base (Figs. 1-2).” (lines 213-215)

20. LINE 192: This sentence is redundant with a sentence earlier in the manuscript on line 152. I suggest removing one of the sentences. If you keep this sentence, then I have the same comment here as mentioned above for line 152.

We chose to keep this sentence so that the thickness information from which we derived the volume estimates are readily available to the reader. But, we did rephrase this text as follows:

“Cerealia Facula ranges in thickness from <3 m at the southern side, to ~5.5 m or ~31 m at the northern side, to ≥ 50 m on top of Cerealia Tholus. Pasola Facula is >6 m thick. Vinalia Faculae have a consistent thickness of only ~2-3 m (Methods 3-4) (Fig. 1b).” (lines 233-235)

21. LINE 201: A subject is needed after the word “this”.

We have rephrased this sentence as follows: “The lack of partially infilled impact craters in Cerealia Facula is consistent with Vinalia Faculae being comparatively thinner and more brine limited...” (lines 243-244)

22. LINES 209-212: Very fascinating results! Thank you.

23. LINE 228: A subject is need after the word “this”. I have the same comment for Lines 251, 450, and 477.

We have rephrased these sentences as follows: “Consequently, the deeper source of Vinalia Faculae is consistent with ...” (line 275); “Crater counts suggest that there could be approximately a few million year age difference between Vinalia Faculae and Cerealia Facula(36), but a few millions of years is unlikely to be a sufficient duration...” (lines 530-532) and “However, even if the lobate materials are as young as ~1 Ma, approximately one million years is still somewhat older...” (lines 558-559)

24. LINE 294: Include a reference for ESRI ArcMap/ArcScene.

We have added the following reference: Whitmeyer, S. J., et al. The digital revolution in geologic mapping. GSA Today, 20 (4-5), 4-10 (2010) (line 369)

25. LINE 298: Your technique using 3D view to more accurately place contacts and stratigraphic relations is neat!

Thank you.

26. LINE 225: What indicates that the brine reservoir would be deeper than the Cerealia and Pasola-Facula-forming brines. Although the Cerealia and Pasola Faculae are sourced from an impact induced melt chamber, couldn't the Vinalia Faculae brine reservoir also be shallow? Some explanation is needed here and/or a reference.

The topographic analyses and thermal modeling predict that the brine layer would be located at the base of Ceres' crust. We have added this clarification into the earlier paragraph: "Instead, Vinalia Faculae could be sourced from a deep, long-lived brine reservoir, which has been suggested to be present at the base of the crust on the basis of topographic analyses(43) and is supported by thermal modeling(44)." (lines 261-263)

27. LINE 303-306: I don't think the last sentence of this paragraph is needed. I suggest removing it.

We have found that there is sometimes confusion in the community about the difference between a geologic map published in a peer-reviewed journal and a USGS SIM. We have left this sentence here so that it is clear this geologic map is not a USGS SIM.

28. LINE 310: I suggest replacing "shortly" with the actual timing of when this event took place. By shortly do you mean days? Years? Thousands of years?

We have rephrased this sentence as follows: "The lobate material was emplaced as a slurry of impact-melted water, salts in solution and blocks of unmelted silicates and salts flowed around the crater interior within a few 1,000s-10,000s years (based on a one-dimensional heat conduction model by L. C. Quick) after Occator's formation(13)." (lines 383-386)

29. LINE 311: The words "comparatively" is used, but it's unclear what comparatively is referring to. Is the lobate material water ice rich compared to the surrounding terrain? Or, to the compositions of the original flows around the crater interior? I suggest rephrasing the sentence to clarify this.

We have rephrased this sentence as follows: "The now-solidified lobate material is comparatively rich in water ice when compared to the surrounding terrain(18), and is covered by..." (lines 79-81)

30. LINE 616: I have one very minor suggestion for Figure 1b. The words “Material” and “here” are not needed. If you’d like to reduce the amount of annotation on your map, you could remove these words in each of the white boxes and include a short explanation in the figure caption.

We made this change and now Figure 1b looks a lot neater.

31. LINE 622: In Figure 2, elevation values are needed on the contour lines.

We have added elevation values to the contour lines in Figure 2.

32. LINE 626: The word correlation should be replaced with spatial relationship or something similar since statistical tests were not performed to quantify a correlation.

We have changed ‘correlation’ to ‘relationship’. (line 766)

33. LINE 628: Although it is mentioned in the figure caption, it would be so much easier to interpret the figure if a legend was provided for the yellow, orange, and white material on Figure 2b.

We have provided this detail in Figure 2b.

34. LINE 650: Figure 4 looks great, but an arrow is needed that points to the landslide in the bottom right of the figure because it’s not immediately obvious. Also, this is very minor, but it’s hard to see the white box around the vent-like structure. I suggest changing the box color. We added the arrow and changed the color of the box to black.

Reviewers' comments:

Reviewer #1 (Remarks to the Author):

I enjoyed reading this revised manuscript, which is much improved. I find that the authors have addressed most of my previous comments, except one. Please see the annotated manuscript where I have commented on the use of the term "melt chamber". As noted, I find this term incorrect, or at best confusing, and would encourage the authors to describe this zone more like the Bowling et al. paper did, as a reservoir of volatiles, but which also contains solid material.

Reviewer #2 (Remarks to the Author):

I am very happy with the revisions made in response to my and the other reviewers' comments. The manuscript is in good shape in every regard. Just a minor formatting thing, on lines 237-238 of the integrated manuscript, the exponents 3 (two mentions) are in very small font size.

The figures are spectacular, and I definitely enjoyed the higher resolution figures. Ambiguities about impact vs endogenic craters are still there, but I am satisfied that the authors have done their level best to discriminate the two types of craters or depressions; and furthermore, the authors acknowledge this ambiguity or possibility of misidentification.

I am happy with the paper as is and I recommend acceptance.

Reviewer #3 (Remarks to the Author):

Dear Authors,

Thank you for addressing my comments and suggestions. I'm happy with the changes made and the revised version of the manuscript. Below I've included the references that you requested regarding compression and material strength of ice:

Gold, L.W., 1977. Engineering properties of freshwater ice. *Journal of Glaciology*, 19, 197-212.

Hobbs, P.V., 1974. *Ice Physics*. Oxford University Press, London.

Haynes, F.D., 1978. Effect of Temperature on the Strength of Snow-Ice, CRREL Report 78-27. U.S. Army Cold Regions Research and Engineering Laboratory, Hanover, N.H.

Chloe Beddingfield

Response to Reviewers (Second Revision) for Scully et al. “The Varied Sources of Faculae-Forming Brines in Ceres’ Occator Crater, Emplaced via Brine Effusion in a Hydrothermal System” (NCOMMS-19-23926A)

Reviewers’ comments are in black text in this response to reviewers document.

Author responses are in blue text in this response to reviewers document.

Changes to the manuscript are in blue text in the revised manuscript.

Response to Reviewer #1:

General Comments:

I enjoyed reading this revised manuscript, which is much improved. I find that the authors have addressed most of my previous comments, except one. Please see the annotated manuscript where I have commented on the use of the term "melt chamber". As noted, I find this term incorrect, or at best confusing, and would encourage the authors to describe this zone more like the Bowling et al. paper did, as a reservoir of volatiles, but which also contains solid material. Thank you for taking the time to review the revised manuscript, and please see our response to detailed comment #5.

Details Comments (from annotated PDF):

1. Line 79: no need for hyphen.
Hyphen removed (line 79).
2. Line 111: from endogenic activity or impact-generated. Would be helpful to clarify this point.
In the Schenk et al. (2019) paper, most of the hydrothermal deposits that are discussed are not impact generated. Thus, we have edited the text as follows: "...previous work found that the morphology of Cerealia Facula is generally consistent with terrestrial, mostly non-impact-generated, hydrothermal deposits(14).” (lines 109-111)
3. Line 123: I like this clarification but this also raises another point. I actually did a search for definitions of "hydrothermal" and there doesn't seem to be consistency; many definitions include the action of only "heated" water, which could be well below zero oC if salinities are high enough. An interesting "analogue" could be the "cold springs" of Axel Heiberg Island in

the Canadian Arctic that are either a few degrees or below zero oC, but that flow year round and through 600 m of permafrost. See 1. Battler M. M., Osinski G. R., and Banerjee N. R. 2013. Mineralogy of saline perennial cold springs on Axel Heiberg Island, Nunavut, Canada and implications for spring deposits on Mars. *Icarus* 224:364–381. and refs therein. This is a very interesting site. We have edited the text as follows, to address this point: “We note that hydrothermal systems do not have to be at or hotter than the boiling point of water: terrestrial hydrothermal springs occur at ambient temperatures(28). Moreover, salts are precipitated from cold springs in the Canadian Arctic that are around or below 0 °C(29).” (lines 122-125)

4. Line 125: which are what? Please add for the non-Cerean experts out there.

We have edited the text as follows to address this comment: “Below the skin depth (ms to cms) at the equator, the average temperature is ~155 K(30), and slightly less at Occator, ~150 K.” (lines 127-128)

5. Line 248: Having read the response to my previous comments on this, I'm afraid that I still find the term "impact-induced melt chamber" incorrect, or at best confusing. The authors refer to the orange zone in figures 1 and 2 of ref 26. Those authors describe this zone as "This reservoir of hydrothermally viable material beneath the crater is composed of a mixture of impactor material and material uplifted from 10's of kilometers beneath the pre-impact surface, which could sample a deep subsurface volatile reservoir, if present." By terming this zone a "melt chamber", whether the authors intend it or not, I think most readers will automatically think of a magma chamber, i.e., a chamber that contains more or less pure melt/magma. My reading and understanding of ref 26 is that this is NOT what is envisaged here. Rather, it is a zone in the crater center that is a mix of solid material (uplifted Cerean crust plus perhaps impactor material) plus H₂O, that now finds itself above the melting point. To me this is more like the Earth's mantle where you have solid (hot) rock and melt that is moving and combining during passage upwards through an interconnected network of fractures.

I would strongly encourage the authors to remove the term "melt-chamber" and instead describe this zone as a "reservoir" (as in ref 26) or something similar.

The descriptor ‘impact-induced cryomagma/melt chamber’ was introduced in the 2018 Hesse and Castillo-Rogez paper to describe this central, hot, volatile-rich region, which would contain significant volumes of melted material: for example, up to ~10,000 km³, because of

the relatively large fraction (≤ 25 vol.%) of water ice in Ceres' crust (Fu et al., 2017). This descriptor has since been adopted by many of the papers in this special issue, e.g. Raymond et al. Moreover, the descriptor clearly differentiates this region from the deep brine reservoir. Thus, we are reluctant to change this descriptor to 'shallow reservoir', or similar, because it would be inconsistent with the preceding/contemporary papers, and could be confused with the deep brine reservoir. However, to address this comment we: (1) have added a caveat that not the entire impact-induced melt chamber would be molten and (2) we explain that the solid fraction would increase over time: "Some solid material (e.g. silicates, which the impact would not be hot enough to melt(27)) would likely be mixed into this melt chamber and, over time, solidification would increase the solid fraction of the impact-induced melt chamber(27,44)." (lines 259-261)

6. Line 270: this seems a bit repetitive as you've essentially already said this at the end of the first paragraph in this section.

We have rephrased this sentence to make it less repetitive: "The impact-induced melt chamber, which feeds the central faculae, is predicted to extend to much shallower depths than the deep brine reservoir(15,25,27,44,46) (Fig. 5)." (lines 276-277)

Response to Reviewer #2

General Comments:

I am very happy with the revisions made in response to my and the other reviewers' comments. The manuscript is in good shape in every regard. Just a minor formatting thing, on lines 237-238 of the integrated manuscript, the exponents 3 (two mentions) are in very small font size. The figures are spectacular, and I definitely enjoyed the higher resolution figures. Ambiguities about impact vs endogenic craters are still there, but I am satisfied that the authors have done their level best to discriminate the two types of craters or depressions; and furthermore, the authors acknowledge this ambiguity or possibility of misidentification. I am happy with the paper as is and I recommend acceptance.

Thank you for taking the time to review the revised manuscript. We have changed the font of the exponents from Times to Arial, which makes the exponents larger for the same font size. (lines 240-241) We have also added the following explanatory text as a result of the peer review process: "Nevertheless, the difficulty in definitively identifying all impact craters from endogenic pits

will likely contribute some unquantifiable errors to the derivation of model ages by the studies discussed in Methods 6.” (lines 488-490)

Response to Reviewer #3: Chloe Beddingfield

General Comments:

Dear Authors,

Thank you for addressing my comments and suggestions. I'm happy with the changes made and the revised version of the manuscript. Below I've included the references that you requested regarding compression and material strength of ice:

Gold, L.W., 1977. Engineering properties of freshwater ice. *Journal of Glaciology*, 19, 197-212.

Hobbs, P.V., 1974. *Ice Physics*. Oxford University Press, London.

Haynes, F.D., 1978. Effect of Temperature on the Strength of Snow-Ice, CRREL Report 78-27.

U.S. Army Cold Regions Research and Engineering Laboratory, Hanover, N.H.

Chloe Beddingfield

Thank you for taking the time to review the revised manuscript and for providing the references.

We have added them into the manuscript as references #39-#41.